# Adjusting for age improves identification of gut microbiome alterations in multiple diseases

Tarini S Ghosh[1], Mrinmoy Das[1,2], Ian B Jeffery[1], Paul W O'Toole[1,2]*

[1]APC Microbiome Ireland, University College Cork, Cork, Ireland; [2]School of Microbiology, University College Cork, Cork, Ireland

**Abstract** Interaction between disease-microbiome associations and ageing has not been explored in detail. Here, using age/region-matched sub-sets, we analysed the gut microbiome differences across five major diseases in a multi-cohort dataset constituting more than 2500 individuals from 20 to 89 years old. We show that disease-microbiome associations display specific age-centric trends. Ageing-associated microbiome alterations towards a disease-like configuration occur in colorectal cancer patients, thereby masking disease signatures. We identified a microbiome disease response shared across multiple diseases in elderly subjects that is distinct from that in young/middle-aged individuals, but also a novel set of taxa consistently gained in disease across all age groups. A subset of these taxa was associated with increased frailty in subjects from the ELDERMET cohort. The relevant taxa differentially encode specific functions that are known to have disease associations.

## Introduction

Alterations in the gut microbiome have been reported for many diseases (*Duvallet et al., 2017*; *Pasolli et al., 2016*). Identifying the specificity of these alterations is necessary for developing microbiome-based diagnostic and therapeutic strategies. In addition to investigating the aspect of causality, three key aspects of microbiome-disease alteration need to be clarified (*Duvallet et al., 2017*; *He et al., 2018*). The first aspect involves the disease-associated components (key taxa and their metabolic functions) which would help in establishing diagnostics and understanding the mechanisms by which the altered microbiome affects the host. The second aspect concerns the directionality of the associations which is required for designing therapeutic strategies (e.g. targeted antimicrobials versus microbiome restoration). The third aspect is the extent of shared versus disease-specific microbiome alterations that indicate what effect the onset of a given disease may have on the microbiome-related risk of other diseases. Meta-analyses across cases and controls from different geographical locations have identified the components of the microbiome alterations common to multiple diseases, as well as disease-specific alterations (*Duvallet et al., 2017*; *He et al., 2018*; *Jackson et al., 2018*; *Pasolli et al., 2016*). While certain diseases like colorectal cancer (CRC) are characterized by an increase (or gain) of pathobionts (such as *Fusobacterium*, *Porphyromonas*, *Parvimonas*), the onset of others like Inflammatory Bowel Disease (IBD) is associated with the depletion of specific taxa (e.g. *Roseburia*, *Faecalibacterium*) (*Duvallet et al., 2017*). In contrast, diarrhoeal diseases are accompanied by both an increase of pathobionts (specifically *Enterobacteriaceae*) as well as lower abundance of commensal taxa. Recent meta-analyses have also shown a surprising degree of overlap of microbiome associations (both at the level of specific taxa as well as specific microbial metabolic pathways) (*Armour et al., 2019*; *Duvallet et al., 2017*; *Pasolli et al., 2016*). Despite these overlaps, links between the microbial taxa and the functional pathways that are consistently altered across multiple diseases have not been explored in detail.

*For correspondence:
pwotoole@ucc.ie

Competing interests: The authors declare that no competing interests exist.

**eLife digest** The human body is an ecosystem made up of both human cells and trillions of microbes, and the largest microbial community is in the gut. This community of gut microbes helps harvest nutrients from our food, modulates our immune system, and even affects our mood. Infectious and chronic diseases appear to cause changes in the make-up of the gut microbiome, while microbiome changes may increase the risk of some non-infectious diseases. Learning more about these disease-linked changes in the gut microbiome may therefore help scientists to develop new tests and treatments. To do this, scientists need to understand which microbes play a role in individual diseases, if risk-related microbes are gained or helpful microbes lost in patients with particular diseases, and if certain changes in gut microbes occur across many diseases.

Ageing also changes the gut microbes. This may happen because older individuals eat a less complex diet and are likely to take many medications that may alter the microbes in their gut. Because of this, age may affect changes in gut microbes associated with diseases. This highlights the need for studies that tease apart the importance of ageing-related and disease-related changes in the gut microbiome.

Now, Ghosh et al. show that gut microbe changes linked to diseases may vary with a person's age. The analysis compared the gut microbiomes of more than 2,500 individuals aged 20 to 89. This included individuals with inflammatory bowel disease, colorectal cancer, type 2 diabetes, intestinal polyps and liver cirrhosis. The study revealed that younger people gradually gain disease-associated gut microbes, while older people tend to lose the gut microbes usually found in a healthy gut. Ghosh et al. also identified a set of gut microbes that were gained in many diseases and across age-groups. This set of microbes was also associated with frailty in elderly people. The characteristics of the microbes in this set are all known to have detrimental effects on human health.

This analysis shows how important it is to control for age and other factors that may skew the results of microbiome projects. Future studies are needed to understand why these gut microbe changes occur and what the consequences of these changes are for a person's health and the course of their disease. This may lead to the development of treatment strategies that help promote a healthy gut microbiome and fight disease throughout life.

Microbiome-based diagnostics/therapeutics also need to account for several host factors. Variations in region/ethnicity of subjects are linked to the gut microbiome composition and microbiome-based diagnostics in various diseases (*Deschasaux et al., 2018*; *He et al., 2018*). In addition to region/ethnicity-specific variations, multiple studies have also shown age to be a strong covariate of microbiome composition (*Falony et al., 2016*). We, along with others, have previously identified ageing-associated microbiome alterations in the ELDERMET cohort, associated with lower complexity dietary intake and poly-pharmacy (*Claesson et al., 2012*; *O'Toole and Jeffery, 2015*; *Ticinesi et al., 2017*). Besides ageing-associated changes in the microbiome, early and late-onset variants of different diseases are characterized by distinct patho-physiologies (*Duricova et al., 2014*; *Yeo et al., 2017*), which could also be linked to age-related microbiome involvement. These observations motivated us to perform an extensive investigation of microbiome-disease alterations across the human age landscape. We aimed to answer the following questions: To what extent does ageing influence microbiome-disease associations for different diseases? Do different diseases show differential microbiome-signatures (both in terms of the microbiome components as well as the directionality of the changes) across age-groups? If so, to what extent do these changes affect the known list of microbiome-based markers for the different diseases? Is there a change in the pattern of multiple disease-associated taxa (as identified by previous meta-analyses) across age groups? And lastly, is it possible to identify a link between these disease markers and the microbial metabolic pathways previously associated with these diseases?

Investigating microbiome alterations in multiple diseases across a wide age-range requires a comprehensive dataset of consistently collated microbiome profiles. The curated MetagenomicData repository of the ExperimentHub R library contains taxonomic and functional pathway profiles of more than 6000 shotgun-sequenced human microbiome samples from different human body sites, spanning more than 30 diseases (and controls) from more than 25 different studies (*Pasolli et al.,*

*2017*). A further advantage of the curatedMetagenomicData is that these microbiome profiles were created using uniform bioinformatic analysis of the sequenced reads (using metaphlan2 and humann2) (*Franzosa et al., 2018*; *Truong et al., 2015*). The repository also provides extensive metadata information for the samples including the study, experimental protocols, region (or country), disease status, age, gender, BMI and antibiotic usage. Although other key factors affecting gut microbiome composition like diet or medication are not provided for all studies, the repository can still facilitate meta-analysis of datasets from multiple studies, to provide initial insights into the effect of other key host associated factors on microbiome disease signatures. Here we analysed the gut microbiome in 15 studies encompassing four diseases, comprising more than 2500 individuals ranging from 20 to 89 years of age, derived from the ExperimentHub repository, the ELDERMET project and recently published studies on IBD) and colorectal cancer CRC (*Franzosa et al., 2019*; *Thomas et al., 2019*; *Wirbel et al., 2019*).

## Results

### Influence of age on the microbiome and microbiome-disease signatures

We first adopted a stepwise methodology to reduce the confounding effects of DNA sequencing/extraction methodologies on the microbiome profiles from the different studies (See Materials and methods; *Figure 1—figure supplement 1*), and subsequently retained samples from individuals with age in the range of 20–89 years (excluding cohorts where the 'controls' also consisted of hospitalized patients) (*Vincent et al., 2016*). We supplemented this data set with 475 shotgun metagenome profiles from recently published studies on IBD (*Franzosa et al., 2019*) and CRC (as 'Validation' cohorts) (*Thomas et al., 2019*; *Wirbel et al., 2019*), finally assembling a collated set of microbiome taxonomic profiles from more than 2500 samples (including the 189 ELDERMET samples used later in the study) (see Materials and methods; *Supplementary file 1* and *Supplementary file 2*; *Figure 1—figure supplement 2*).

We next investigated the interaction of metadata with taxonomic profiles. After filtering out redundant and sparse metadata types (recorded for less than 30% of the samples), we performed PERMANOVA linking the effects of each metadata with the gut microbiome accounting for DNA extraction technique as a confounder (See Materials and methods; DNA extraction technique was still treated as a confounder as it still had a minor effect, $R^2 = 0.019$, on the gut microbiome composition). Regional factors namely country and continent had the largest interaction with gut microbiome composition (*Figure 1A*). Regional factors reflect the ethnicity and other socio-economic properties of the study populations, which has a dominant effect on gut microbiome architecture and microbiome-based disease signatures (*Deschasaux et al., 2018*; *He et al., 2018*). Age followed by study-condition (disease versus control status) were the second major effect. We thus explored the variation of the apparently 'healthy' microbiome across the age landscape using Principal Component Analysis of 1175 gut microbiome profiles from exclusively 'control' individuals from the age-groups' 20–39' (categorized as 'Young'),' 40–59' ('Middle-Age') and' above 60' ('Elderly'). PERMANOVA analysis of the gut microbiome composition (after adjusting for country and the DNA extraction technique) indicated that the elderly controls had a significantly distinct microbiome composition compared to the young and the middle-aged (p<0.001) (*Figure 1B*), in line with our previous findings (*Claesson et al., 2012*). To further explore this effect, we then investigated the effect of subject age on the gut microbiome composition in five diseases, IBD, type II diabetes (T2D), intestinal polyps, CRC, and liver cirrhosis, for which substantial number of cases were available across at least two age-groups. Samples corresponding to different diseases were available in different cohorts (*Table 1*). Each cohort dataset comprised healthy controls and diseased individuals matched with respect to geographical region and environment (thereby reducing these confounding effects on the gut microbiome), so we were able to perform PERMANOVA to quantify the effect of both age group ('Young', 'Middle-Aged' and 'Elderly') and disease status ('Control' and 'Disease') on the gut microbiome (separately in each of the different disease-specific cohorts). For all five diseases, age had a significant effect. Notably, for three diseases (T2D, CRC and intestinal polyps), this effect was higher than that of the disease itself (*Figure 1C*).

Next, we investigated whether these age-related changes in microbiome composition influenced disease signatures. However, performing this investigation posed a specific challenge. Given that

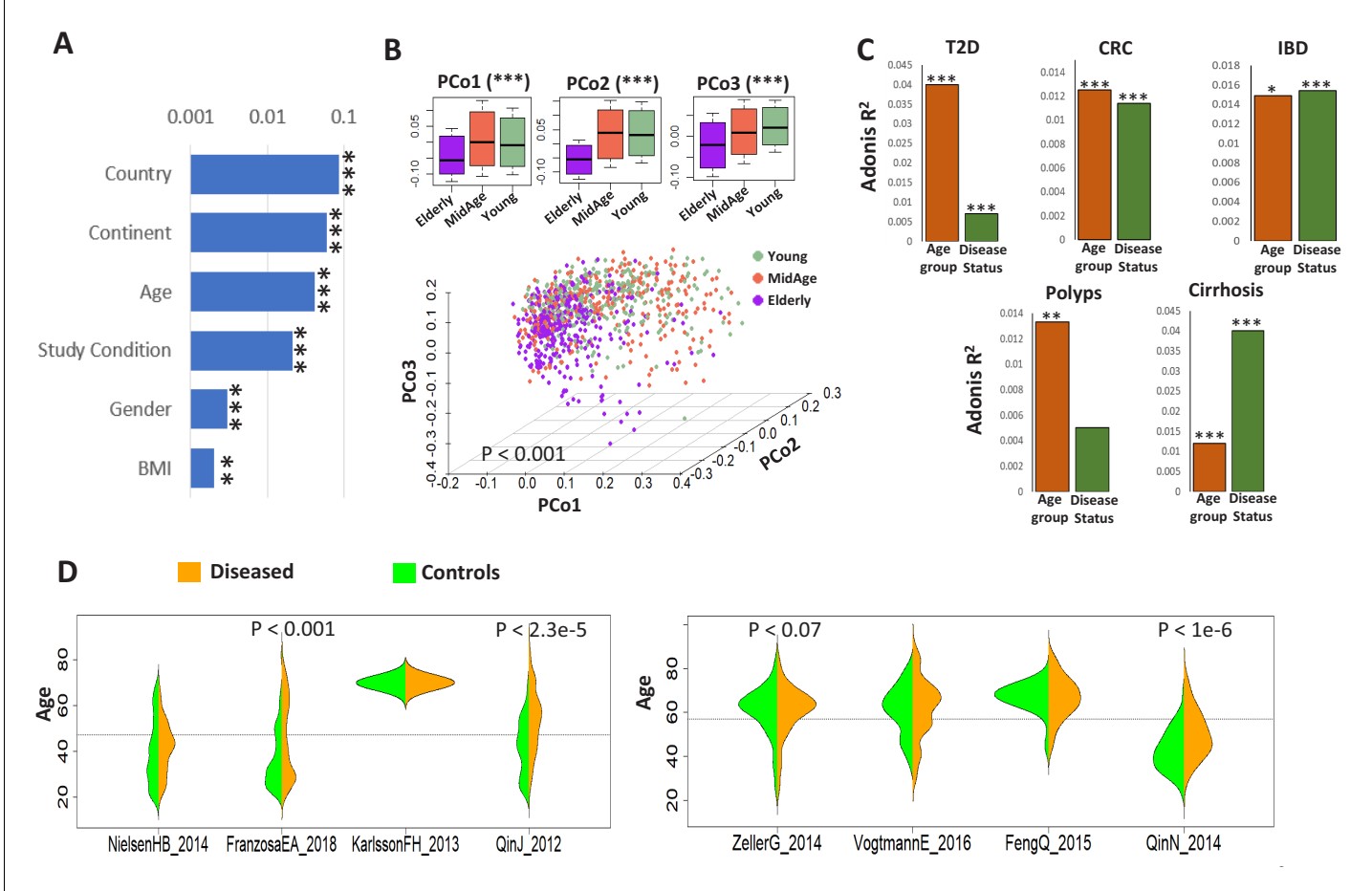

**Figure 1.** Age influences microbiome composition as well as microbiome-disease signatures. (A) Bar plots showing the effect (denoted by $R^2$ values computed using PERMANOVA after adjusting for the DNA extraction technique as the confounder) of host factors with microbiome composition in the ExperimentHub repository. Only metadata available for at least 30% of the samples are shown. The p-values for the significance of association are also indicated as ****: p<0.0001; ***: p<0.001, **: p<0.01, *: p<0.05. (B) Principal Co-ordinate Analysis (PCoA) plots of the species profiles of the 'control' samples grouped into three age ranges, Young (20–39 years), Middle (40–59 years) and Elderly (60 years and above). The significance (p-value) of the differences between the three groups, computed using PERMANOVA (adonis) after considering the country-specific differences and the DNA extraction technique, is also indicated. The boxplots on the top show the variation of the top three PCoA coordinates for the samples belonging to the three age-groups. The elderly harboured a significantly different microbiome compared to the young/middle-aged. (C) Barplots of PERMANOVA $R^2$ values showing the variation of microbiome with disease (adjusting for age-group) and age-group (adjusting for disease status) in the five disease cohorts. The Cohort-specific analyses ensured that the variations observed were not due to country-specific regional differences in microbiome composition. However, within each cohort, there were skews in the representation of diseased and control samples from different age-groups (as seen in *Table 1*). Furthermore, in four out of the eight cohorts, there were significant differences in the age variation of control and diseased individuals, as shown by the beanplots in D.

The online version of this article includes the following figure supplement(s) for figure 1:

**Figure supplement 1.** Effect of median read length and DNA extraction techniques on the microbiome variation.

**Figure supplement 2.** Pictorial summary describing the workflow used for preparing a core set of around 2564 gut metagenomic datasets derived from the publicly available datasets (curatedMetagenomicData[9] and Franzosa et al 2018[8]) and the ELDERMET repository.

**Figure supplement 3.** Number of control and diseased individuals belonging to the different age-groups present in (A) country-specific and (B) continent-specific groups pertaining to each disease.

regional factors like country or continent had the highest effect on the gut microbiome composition, any investigation into age-group-specific microbiome signatures for the various diseases required the comparisons to be performed within geographically homogeneous sub-populations (or study cohorts) to ensure that any differences in disease signatures were not simply driven by regional variations in gut microbiome composition. However, focusing on specific disease cohorts (which belong to specific countries) also imposes limitations including reduction in sample size (and therefore

**Table 1.** Number of control and diseased individuals belonging to the different age-groups present in the continent-specific groups pertaining to each disease.

Age-groups where the number of control/diseased samples are less than 15 are highlighted in red. The shortened notations for the different country used are ESP: Spain; USA: United States, CHN: China, SWE: Sweden, AUT: Austria, FRA: France.

| CRC Cohorts | Country | Young | | Middle | | Elderly | |
|---|---|---|---|---|---|---|---|
| | | Control | Disease | Control | Disease | Control | Disease |
| ZellerG_2014 | FRA | 5 | 0 | 15 | 14 | 33 | 31 |
| FengQ_2015 | AUT | 0 | 0 | 4 | 10 | 57 | 36 |
| VogtmannE_2016 | USA | 2 | 3 | 17 | 18 | 41 | 39 |

| Cirrhosis Cohorts | Country | Young | | Middle | | Elderly | |
|---|---|---|---|---|---|---|---|
| | | Control | Disease | Control | Disease | Control | Disease |
| QinN_2014 | CHN | 64 | 19 | 45 | 77 | 5 | 26 |

| IBD | Country | Young | | Middle | | Elderly | |
|---|---|---|---|---|---|---|---|
| | | Control | Disease | Control | Disease | Control | Disease |
| FranzosaCA_2018 | USA | 36 | 77 | 16 | 41 | 45 | 14 |
| NielsenHB_2014 | ESP | 35 | 53 | 22 | 84 | 12 | 8 |

| T2D Cohorts | Country | Young | | Middle | | Elderly | |
|---|---|---|---|---|---|---|---|
| | | Control | Disease | Control | Disease | Control | Disease |
| KarlssonFH_2013 | SWE | 0 | 0 | 0 | 0 | 43 | 53 |
| QinJ_2012 | CHN | 69 | 27 | 89 | 85 | 10 | 57 |

| Polyps Cohorts | Country | Young | | Middle | | Elderly | |
|---|---|---|---|---|---|---|---|
| | | Control | Disease | Control | Disease | Control | Disease |
| Feng_2015 | AUT | 0 | 0 | 4 | 8 | 4 | 8 |
| ZellerG_2014 | FRA | 5 | 0 | 15 | 13 | 41 | 29 |

statistical power) and bias for certain age-groups. Among the eight disease cohorts (in the curatedMetagenomicData) that we analyzed here, we discovered that half of these featured a significant difference in the age of the control and diseased individuals (i.e. they were not age matched) (*Figure 1D*). In addition, there were skews in the representation of specific age-groups for most of the disease cohorts. Adding 'control' samples from other cohorts that were collected from the same region (country or continent) as the corresponding disease cohorts was expected to circumvent this issue by increasing the number of available samples and thereby the statistical power of comparisons. We addressed this issue by grouping the samples into disease-specific country-/continent-level bins (See Materials and methods; *Figure 1—figure supplement 3*). This ensured sufficient representation of samples from the three different age-groups for all diseases.

Next, we performed investigations using two different approaches to explore the effect of age on disease-microbiome signatures. In the first analysis, within each disease-specific country-level cohort, we performed PERMANOVA investigating the effect of the interaction between disease signatures and age-group, after adjusting for the effects of country (given that there were samples from different countries within the different disease cohorts) and the independent effects of disease and age-group (*Table 2*). Interaction between two metadata categories measures the extent to which the microbiome variation with respect one metadata (in this case, the disease) is influenced by the variation in the other (that is the age-group). For three of the diseases (IBD, CRC and Polyps), the effect of the interaction of disease with age-group was significant (all with $p<0.05$; PERMANOVA after adjusting for the confounders as above). The effect was also marginally significant for T2D

**Table 2.** Results of PERMANOVA analysis investigating the effect of the interaction between disease signatures and age-group, after adjusting for the effects of country (within the continent cohorts) and the independent effects of disease and age-group.

| Adonis | T2D | | | IBD | | | CRC | | | Polyps | | | Cirrhosis | | |
|---|---|---|---|---|---|---|---|---|---|---|---|---|---|---|---|
| | F. Model | R-Squared | P-Value | F. Model | R-Squared | P-Value | F. Model | R-Squared | P-Value | F. Model | R-Squared | P-Value | F. Model | R-Squared | P-Value |
| Country | 16.69 | 0.029 | 0.001 | 11.34 | 0.017 | 0.001 | 7.63 | 0.027 | 0.001 | 4.65 | 0.022 | 0.001 | NA | NA | NA |
| Disease | 4.81 | 0.008 | 0.001 | 15.79 | 0.023 | 0.001 | 4.70 | 0.008 | 0.001 | 1.25 | 0.006 | 0.028 | 12.107 | 0.029 | 0.001 |
| Age-Group | 1.38 | 0.005 | 0.001 | 2.51 | 0.007 | 0.001 | 3.80 | 0.007 | 0.001 | 1.00 | 0.004 | 0.408 | 1.239 | 0.006 | 0.016 |
| Disease: Age-Group | 1.12 | 0.004 | 0.08 | 2.73 | 0.008 | 0.001 | 3.57 | 0.006 | 0.001 | 1.38 | 0.006 | 0.011 | 1.035 | 0.005 | 0.290 |

(p<0.08; PERMANOVA). This indicated that (even after considering the regional variations and the individual influences of disease and age), for these diseases, the microbiome variation associated with disease was significantly influenced by the variations in the age-group of the individuals. Further investigation of these influences (measured by the $R^2$) also indicated that, while IBD and CRC had the highest influence of age as a covariate of disease signatures, with T2D and Polyps relatively lower but significant influences (*Table 2*).

In the second analysis, we utilized Random Forest (RF) Classifier Models. Random Forest models belong to a category of ensemble-based machine learning methods (based on decision trees), that can be used to predict a given trait (in this case, disease status) based on a list of predictor features. Random Forests can not only be used to judge the strength of the association between the predictors and the response (based on the accuracy or area under the curve (AUC) of prediction), but also whether associations identified on one set of observations are extrapolatable to another. Consequently, Random Forest models have been routinely used in microbiome based studies to not only quantify and characterize the microbiome associations with various diseases (*Feng et al., 2015*; *Karlsson et al., 2013*; *Pasolli et al., 2016*; *Qin et al., 2012*), but also to study the transferability of the associations predicted in set of individuals on others (*He et al., 2018*; *Thomas et al., 2019*).

We specifically focused on the disease-specific country-level bins (to ensure regional homogeneity as much as possible). Subsequently, we divided subjects (in the country level cohorts corresponding to each disease) into three age bands (as described above) and performed 100 iterations, each time training Random Forest classifiers on a subset of samples belonging to an age band, and evaluating the disease classification performance on the samples from the same age band (excluding the samples used for training) (Same Age-group classification) or the other age bands (Different Age-group classification) (while keeping the training and testing data set sizes constant across age-bands) (see *Figure 2—figure supplement 1* and Materials and methods for details). In summary, in each disease-age-group scenario, we used two different approaches to assess whether disease classification performance was significantly different between Same Age-group classification and Different Age-group classification. In the first approach, for the 100 iterative re-sampled classifier models, we compared the AUCs obtained for the Same age-group classification with those obtained for the Different Age-group classification using paired Wilcoxon Signed Rank tests. In the second approach, using Permutation tests, for the 100 iterative re-sampled classifier models, we compared if the actual difference in AUC for the age-group cohorts (Same Age-group classification – Different Age-group classification) is significantly different from what would be expected by random (null distribution generated by permuting the age-group labels of the subjects) (*Figure 2—figure supplement 1* and Materials and methods).

The data revealed large differences across age groups for the various diseases. Specifically, in 10 of the 13 disease-age group scenarios (covering five diseases), classifiers trained and tested on the same age-group had significantly higher disease prediction AUC than when tested on different age-groups (*Figure 2*), with the improvement of classification performance significantly higher than would be expected by random chance (obtained using the permutation test based strategy) (*Figure 2—figure supplement 2*). This confirmed that microbiome-disease associations had age-centric trends. Furthermore, the extent of the differences was pronounced for IBD, CRC and T2D (the above

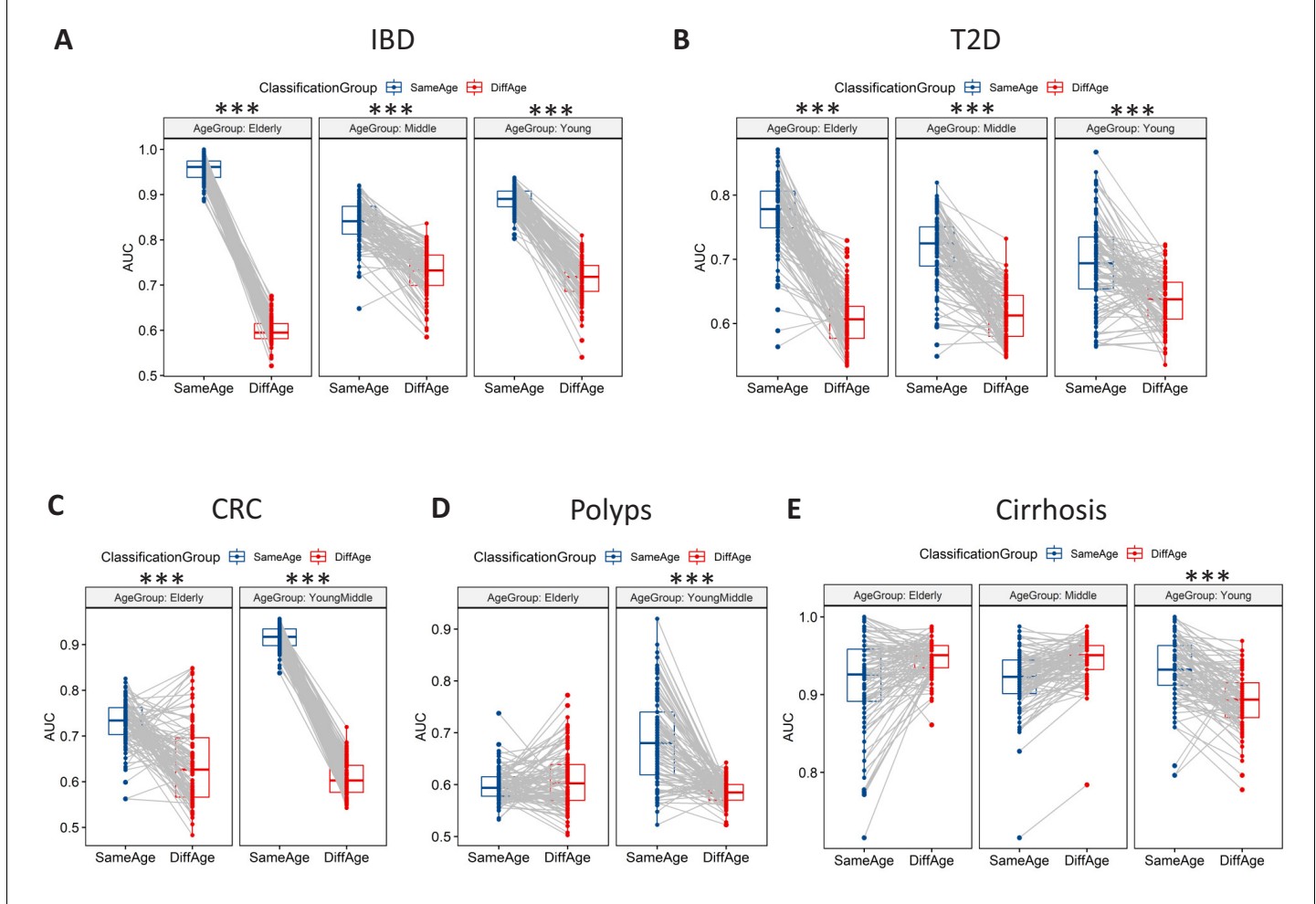

**Figure 2.** Microbiome-disease signatures display specific age group centric trends. Boxplots showing the variation of disease-classification area under the curve (AUCs) when classifiers trained on one age-group were tested on either the same (denoted as SameAge or Same Age-group classification) or different age-groups (denoted as DiffAge or Different Age-group classification) for (**A**) IBD (**B**) T2D (**C**) CRC (**D**) Polyps and (**E**) Cirrhosis. Each point denotes the median AUC (of 20 iterations) obtained using each of the 100 sub-sample based Random Forest classifier models when tested on samples from the Same Age-group (in blue) or Different Age-groups (in red). Median AUC values obtained for the same classifier for Same Age-group and Different Age-group classification are joined by grey lines. Scenarios where in the Same Age-group classification had a significant increase of classification AUC as compared to the Different Age-group are indicated (using the P-values of significance). The Wilcoxon signed rank test p-values of significance, after correction using Holm method, are indicated as ***: p<0.001, **: p<0.01, *: p<0.05.

The online version of this article includes the following source data and figure supplement(s) for figure 2:

**Source data 1.** Number of disease and control samples in different age-groups obtained by collating samples from datasets from the same (**A**) Countries and (**B**) Continents as the disease-specific datasets.

**Figure supplement 1.** Schematic workflow of the methodology adopted for comparing the performance of disease-specific random forest classifiers trained on one age-group when applied to test samples from the same (Same Age-group classification) or different age-groups (Different Age-group classification) using Wilcoxon Signed Rank tests.

**Figure supplement 2.** Boxplots comparing the actual AUC differences (that is, median AUC for same age-group classification – median AUC for the different age-group classification) obtained for classifiers (in each disease-age-group scenario) with the null distribution of AUC differences obtained between two permuted sets (as obtained in the Permutation tests).

trend being reflected across all age-groups), reflecting the patterns observed in the PERMANOVA analysis (*Table 2*). CRC and Polyps were characterized by noticeably similar age-specific trends wherein the elderly age-groups had a noticeable decrease in classification AUCs (that is lower AUCs for classifiers trained or tested on the elderly age-groups), indicating that the microbiome-disease signatures for these diseases in the elderly age-groups are weaker (*Figure 2*). For these two diseases, the AUC difference between the Same Age-group and Different Age-group disease

classification was also reduced in the elderly (in the case of Polyps, this difference was not significantly different what would be expected by random chance) (*Figure 2—figure supplement 2*).

## Age-centric differences in microbiome-disease associations

We next sought to identify differentially associated age-disease markers and how these corresponded with the currently known microbiome-disease associations. Besides quantifying the strength and extrapolatable nature of the microbiome signatures, iterative RF models also provided the list of taxa contributing to the models along with their relative importance scores (for each disease age-group scenario). Therefore, in the first step, based on the RF analysis (as described above) for each disease, we computed the age-group specific feature importance scores for each taxa and then shortlisted the taxa having marker scores in the top 85 percentile individually for each of the age-groups (See Materials and methods; *Figure 3—figure supplement 1*; *Figure 3—source data 1*). Comparing the top 85 percentile markers across the different age-groups revealed that the proportion of age-influenced markers (or markers differentially associated across at least one age-group) ranged from 30% in CRC to 50% in IBD (greater than 35% for the other three) (*Figure 3—source data 1*; *Figure 3—figure supplement 2*). The feature importance scores (obtained using the iterative RF models) were then compared across age-groups and taxa having significant differences in the feature importance scores (across age-group) (Benjamini FDR < 0.01 Kruskal-Wallis test for IBD, T2D and Cirrhosis and Mann-Whitney Tests for CRC and Polyps) were then identified (*Figure 3—source data 1*).

These results indicate that the microbiome associations with disease have both an age-group-specific and an age-group-independent component. However, many of the age-group-specific changes in microbiome-disease association may also be reflections of changes that accompany ageing in general. Age had the second major effect on microbiome composition, and it was important to investigate these age-specific changes in disease-microbiome associations after deconvoluting for the effect of ageing. Secondly, while grouping samples into country-level bins ensured enough regional homogeneity of the compared groups, there were still biases in the representation of certain regions in certain age-groups for the different diseases (*Figure 1—figure supplement 3*). Given that RF models cannot intrinsically adjust for these confounding effects, we probed this using a linear regression-based strategy that compared the extent of influence that age-group had on the disease-association pattern of a taxon (Disease:Age-group) as compared to the individual influences of disease and age-group (see Materials and methods; *Figure 3—figure supplement 3*). We specifically investigated those taxa having significant differences in their feature importance scores (*Figure 3—source data 1*) and identified those having significantly higher influence of Disease interacting with Age-group than Age-group alone (Log likelihood test one sided p<0.05) (*Figure 3—source data 2*; *Figure 3*) as the final validated list of 'strictly age-specific disease markers'. Notably, the percentage of taxa showing significant differences in their feature importance scores that were also validated in the Linear regression-based approach, showed disease-specific trends reflecting those observed in our PERMANOVA-based (*Table 2*) and RF-based analysis (*Figure 2*). Specifically, for IBD and CRC, more than 63% of features showing significant differences in their marker scores across age-groups were also validated in the linear-regression-based validation, indicating that for these diseases, a majority of age-group-specific disease associations persist even after accounting for the overall effect of ageing. In contrast, for Polyps and Cirrhosis these overlaps showed a progressive decrease (*Figure 3—figure supplement 4A*).

We next compared the age-linked disease markers identified here with the disease markers originally reported. For four of the diseases (i.e. not CRC), we compiled a list of known disease-associated markers from each of the original studies (*Feng et al., 2015*; *Franzosa et al., 2019*; *Karlsson et al., 2013*; *Qin et al., 2012*; *Qin et al., 2014*). For CRC, a recently study presented a meta-analysis of all major cohorts (encompassing those in curatedMetagenomicData) (*Feng et al., 2015*; *Vogtmann et al., 2016*; *Zeller et al., 2014*), along with three newly sequenced cohorts (*Thomas et al., 2019*), to produce a refined set of cross-cohort validated CRC markers. We utilized this list to compile the set of known markers for CRC. Comparing the list of these associations with a pre-compiled list of all known marker associations reported in the original studies (*Figure 3—source data 3*) (*Feng et al., 2015*; *Franzosa et al., 2019*; *Karlsson et al., 2013*; *Qin et al., 2012*; *Qin et al., 2014*; *Vogtmann et al., 2016*; *Zeller et al., 2014*), identified that 40% of taxa identified in the original study for Polyps, more than 63% for CRC and IBD, 58% for T2D and 25% for Cirrhosis,

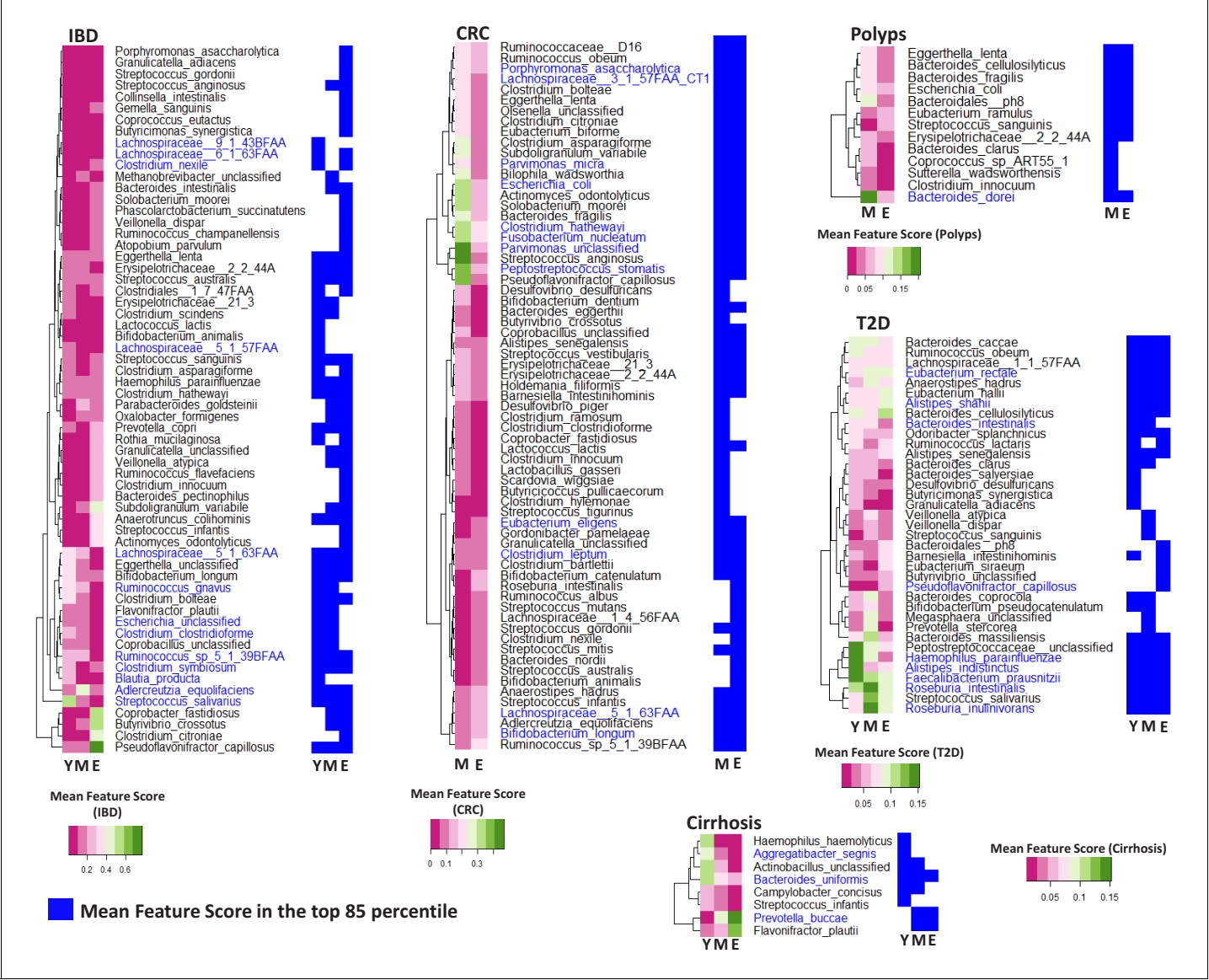

**Figure 3.** Specific taxa show age-group linked trends of disease association. Heatmaps showing the marker scores for the list of taxa that are differentially associated with the indicated disease across the age-groups (Y: Young; M: Middle-aged and E: Elderly). For each disease, this list of species was selected as those which were among the top 85 percentile features in at least one age-group and which displayed significant variation in their feature importance scores across at least two age-groups. These taxa were further validated using a linear regression approach to ensure that their age-group specific association with disease was significant even after accounting for the independent changes associated with ageing. The font colors of the species indicate whether the species were reported in the original studies as being associated with the given disease (Dark blue: Associated Previously; Black: Not Associated). For each disease, heatplots (adjoining on the right side of the corresponding heatmap) shows the different taxa were identified within the top 85 percentile markers for each age-group (in blue color).

The online version of this article includes the following source data and figure supplement(s) for figure 3:

**Source data 1.** Marker scores for the top 85-percentile markers (detected across at least one of the age-groups) for the five diseases, along with the P-values of the comparisons of these scores across age-groups.

**Source data 2.** Linear model based validation results to deconvolute the effect of ageing on age-group specific disease markers for (**A**) IBD (**B**) Cirrhosis (**C**) T2D (**D**) CRC and (**E**) Polyps.

**Source data 3.** Markers identified in the original datasets for (**A**) T2D, (**B**) IBD, (**C**) Cirrhosis, (**D**) CRC, and (**E**) Polyps as either significantly different between the diseased and control cohorts or having discriminatory power for the classification of diseased samples from microbiome composition.

**Figure supplement 1.** Variation of feature importance scores of the taxa across the iterative Random Forest models.

**Figure supplement 2.** The percentage of 85 percentile taxa that were detected as common or specific to certain age-groups for the five different diseases.

*Figure 3 continued on next page*

*Figure 3 continued*

**Figure supplement 3.** Schematic workflow describing the linear regression-based strategy to deconvolute the effect of ageing from age-specific disease association.

**Figure supplement 4.** Validation of age-specific trends using Linear Regression approach and the effect of these trends on the known markers for the various diseases.

displayed significant differences in their feature importance scores across age-groups, thereby emphasizing that age-linked disease-microbiome associations are prevalent across diseases (*Figure 3—figure supplement 4B*). Even in the final list of 'strictly age-specific markers' that were deconvoluted for the effect of ageing, the overlap varied from 20% for T2D and Polyps to as high as 63% for CRC (*Figure 3*; *Figure 3—figure supplement 4C*). The effects were especially pronounced for IBD, where many of the known markers (species belonging to Lachnospiraceae, *Escherichia*, *Clostridium clostridioforme*, *Clostridium nexile*, *Blautia producta*) were not even identified in the top 85 percentile in certain age-groups (*Figure 3*). For CRC, some of the well-known markers *Fusobacterium nucleatum*, *Parvimonas micra*, *Peptostreptococcus stomatis*, and *Porphyromonas asaccharolytica,* although identified in the top 85 percentile across age-groups, the strength of the associations in terms of RF feature importance scores as well as the results of linear modelling showed age-specific trends. The same pattern was also observed for some of the signature taxa for T2D (*Faecalibacterium prausnitzii*, *Roseburia intestinalis*, *Haemophilus parainfluenzae*, *Bacteroides intestinalis* and *Alistipes indistinctus*) and Polyps (*Bacteroides dorei*). The variations in the strength and stability of these associations were further evident when we compared the percentage of times each marker is identified with scores greater than 85 percentile in different age-groups (within the 100 iterations for each age-group) (*Figure 3—figure supplement 4D*). Several known markers of CRC (*Parvimonas micra*, *Eubacterium eligens*) and T2D (*Bacteroides intestinalis*, *Haemophilus parainfluenzae*) are identified in the top 85 percentile across more than 80% iterations in certain age-groups, but less than 50% in others. This has implications for the efficacy of microbiome-directed diagnostic/therapeutic strategies across age groups.

## Replicability of the age-centric microbiome-disease signatures across multiple cohorts in CRC

We next tested if the differential trends of disease association observed were affected by cohort-specific differences. For CRC, we observed the strongest effects of age-group influenced disease signatures which affected more than 63% known CRC markers, and that were validated even after considering country-specific representations and deconvoluting the effect of ageing. Furthermore, the three newly sequenced cohorts (referred to as 'Cohort1', 'Cohort2' and 'WirbelJ_2019', as a group 'Validation Cohort') had sufficient representation of samples from different age-groups (*Supplementary file 1*) (*Thomas et al., 2019*; *Wirbel et al., 2019*), and represented new data that had not been included in our initial identification of differentially associated markers. In the curatedMetagenomicData repository, the CRC disease samples were collected from three cohorts, namely ZellerG_2014, VogtmannE_2016 and FengQ_2015 (*Feng et al., 2015*; *Vogtmann et al., 2016*; *Zeller et al., 2014*). To explore the reproducibility of the associations (irrespective of the general effect of ageing) as well as to understand the biological basis for these differential associations, we validated the age-aware RF classification models designed on this cohort (referred to as the Training Set1) on the validation cohort (*Figure 4A*; upper panel) (using a similar strategies as described in *Figure 2—figure supplement 1*). The age-specific trends of disease classification were similar to those obtained earlier (*Figure 2*), that is, classifiers trained on the young/middle-aged individuals had the highest classification performance when tested on the same age-group, with a significant reduction in classification AUCs when tested on the elderly. In contrast, performance decreased significantly for those either trained or tested on elderly (where the difference between the Same Age-group and Different age-group testing was not observed to be significant) (*Figure 4A*; *Figure 4—figure supplement 1*). We further re-generated similar iterative age-group specific microbiome disease-prediction models by taking subsets of samples from within the 'Validation Cohort' (referred to as Training Set2) and testing on non-training subsets of the same. The classification pattern remained invariant (*Figure 4A*; lower panel versus upper panel). These results indicated that the

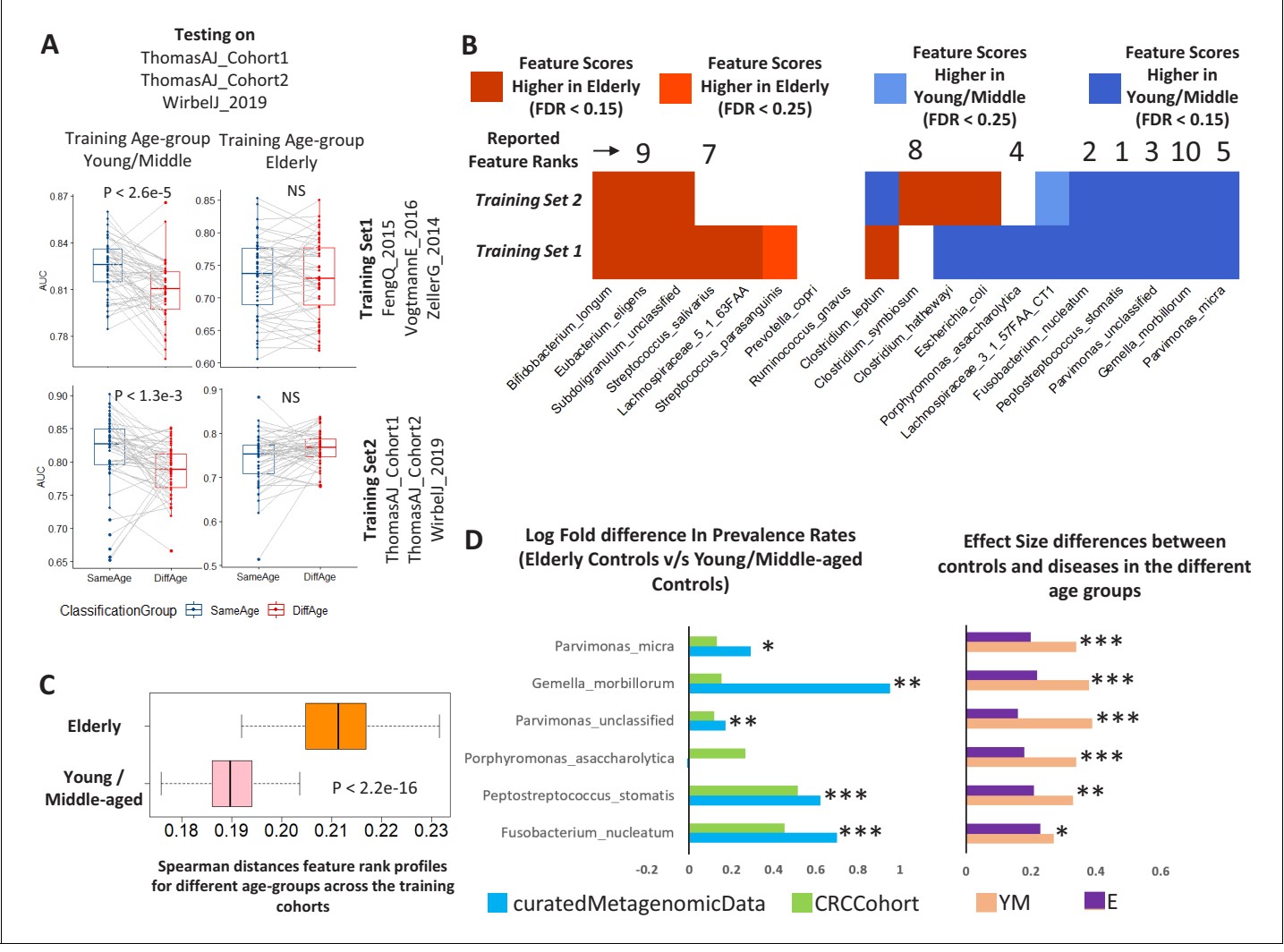

**Figure 4.** Age-dependent CRC-specific markers are reproducible across multiple cohorts and ageing-associated changes make the elderly gut microbiome disease-like. (A) The boxplot on the top panel shows the distribution of AUC values obtained when classifiers trained on different age-groups (YM: Young/Middle-aged; E: Elderly) in three cohorts of the curatedMetagenomicData (Training_Set1: ZellerG_2014, FengQ_2015 and VogtmannE_2016) are tested on the three datasets of the validation cohort (ThomasAJ_Cohort1, ThomasAJ_Cohort2 and WirbelJ_2019). The lower panel shows the same, but with age-group specific classifiers trained from within the validation cohort (Training_Set2). Both the classification models generated the same trends of classification, indicating age-group specific reproducibility of the disease signatures. The description of the point colors is the same as for *Figure 2*. (B) Age-group dependent associations of the known CRC markers in the two independent cohorts, namely Training_Set1 (curatedMetagenomicData) and Training_Set2 (Validation Cohort). Shades of blue indicate higher feature importance scores in the young/middle-aged and red indicates higher feature importance scores in the elderly. FDR p<0.15 indicates features identified as being high either in elderly of young/middle with Benjamini-Hochberg corrected Mann-Whitney test p-value<0.15. FDR p<0.25 indicates features identified as being high either in elderly of young/middle with Mann-Whitney test p-value<0.25. Out of the 19 known and validated CRC-markers (obtained from *Thomas et al., 2019*), 13 showed significant differences in their feature importance scores across the two age-groups (in the curatedMetagenomicData cohorts). For nine of these 13 markers, the pattern of associations could be reproduced in the Validation cohort, further indicating the replicability of the obtained results. The feature ranks of the top 10 markers obtained in *Thomas et al. (2019)* are also shown. Six of the top 10 markers show increased association, but only within the young/middle-aged. Only one of the markers associated with the elderly. This indicates a loss of disease-signature in the elderly. (C) Across cohort Spearman distances of feature rank profiles obtained for the disease classifiers trained on the different age-groups (See Materials and methods). A stable disease signature would result in reproducible species rank profiles across cohort and consequently lower Spearman distances. While this is the case for young/middle-aged, the elderly signatures obtained for the different cohorts show significantly high Spearman distances (showing significant variations and lack of disease signature). (D) The log ratios of the prevalence rates of the top six CRC-associated markers in elderly controls with respect to the young/middle-aged controls (in both the curatedMetagenomicData and CRC-specific cohorts). A positive value indicates higher prevalence rates in elderly controls. The significance of the increase is also indicated (p-values of fishers' exact test combined using Fisher method) as ***: p<0.001, **: p<0.01, *: p<0.05. The increase in the elderly is characterized by a significant decrease in the effect-size differences between the controls and diseased in elderly, leading to masked signatures.

*Figure 4 continued on next page*

*Figure 4 continued*

The online version of this article includes the following figure supplement(s) for figure 4:

**Figure supplement 1.** Results of the permutation test (as described in *Figure 2—figure supplement 1*) applied for the testing of the CRC Validation datasets using the different training cohorts as indicated in the Figure.

---

variations in age-group specific microbiome-disease signatures remained stable and were not derived from biases in the training datasets.

We then examined the overlap of the age-dependent associations of the known CRC markers across the two training models (*Figure 4B*). From among the refined cross-cohort associated subset of 19 CRC-associated taxa (*Thomas et al., 2019*), 13 (67%) had differential association with either of the age-groups (with FDR corrected p-value<0.1) in Training Set 1. Nine of these (77%) could be replicated with the same directionality in the Training Set 2 (*Figure 4B*). Interestingly, from among the top CRC-predictive features reported in the recent meta-analysis, six were found to be associated with higher predictive power only in the young/middle age category (*Figure 4B*). Only one was observed to be associated with the elderly microbiome.

We next checked the stability of the feature-associations of CRC across the young/middle and the elderly age-groups across the cohorts. The bootstrapped approach adopted in this study enabled obtaining the microbiome-disease signatures for multiple subsets of diseased and control subjects. In a scenario where the microbiome signature is robust and stable, the microbiome-disease associations (which in this case are obtained as the feature rank profiles for each of the taxa) from each iteration of classifier should be similar (across the cohorts). However, in this specific case, while the classifier profiles for the young/middle-aged individuals across the two cohorts were relatively similar, the inter-training-cohort distances obtained for the elderly-specific disease classifiers were significantly higher (indicating variable disease signature and loss of reproducibility) (p<2.2e-16; Mann Whitney Test) (*Figure 4C*).

Although no specific trends were observed when comparing abundances of the known CRC-specific markers, five of the top ten markers (*Fusobacterium nucleatum*, *Peptostreptococcus stomatis*, *Gemella morbillorum*, *Parvimonas micra* and *Parvimonas spp*) showed significantly higher prevalence rates in the elderly controls (reproducibly across both datasets) (*Figure 4D*). This resulted in a significant reduction in the effect size of their differences (between control and diseased individuals) in the elderly age-group. Thus, ageing-associated changes may render elderly individuals susceptible to specific microbiome-related diseases like CRC.

## Age-specific changes in the directionality of taxon abundance alterations for specific diseases, and the microbiome response shared by multiple diseases

We next investigated if the diseases were characterized by distinct patterns of microbial taxon gain or loss, even across the different age-groups. We first focused on the individual study cohorts (*Figure 1C*). For each disease-age-group scenario, we determined the directionality (increased versus decreased in disease) of association of the corresponding top disease-predictors by comparing their abundance trends in (study-matched) control and diseased samples (Mann Whitney Test p<0.05). For four of the five diseases (i.e. except polyps), there was a significant change in the directionality of the microbiome alteration, characterized by a significant reduction in the number of gained features with older age (Fishers' exact test p<0.05; *Figure 5—figure supplement 1*). To add statistical power and check whether the above trend is retained across a larger cohort, we performed the analysis using 'continent-matched' disease and controls (using reasonably stringent thresholds of Benjamini-Hochberg corrected P value < 0.1 obtained from Mann Whitney tests). The trend remained similar. For elderly, microbiome alterations (for all diseases except cirrhosis and polyps) were characterized by a significant increase in the number of lost taxa (*Figure 5A*). The list of disease-specific markers with significant directionality of disease association across age-groups are presented in *Figure 5—source data 1*. Thus, the microbiome alterations in most of these diseases are characterized by a gradual shift from a state dominated by gained microbiome components to an increasing loss of control-associated taxa in the elderly.

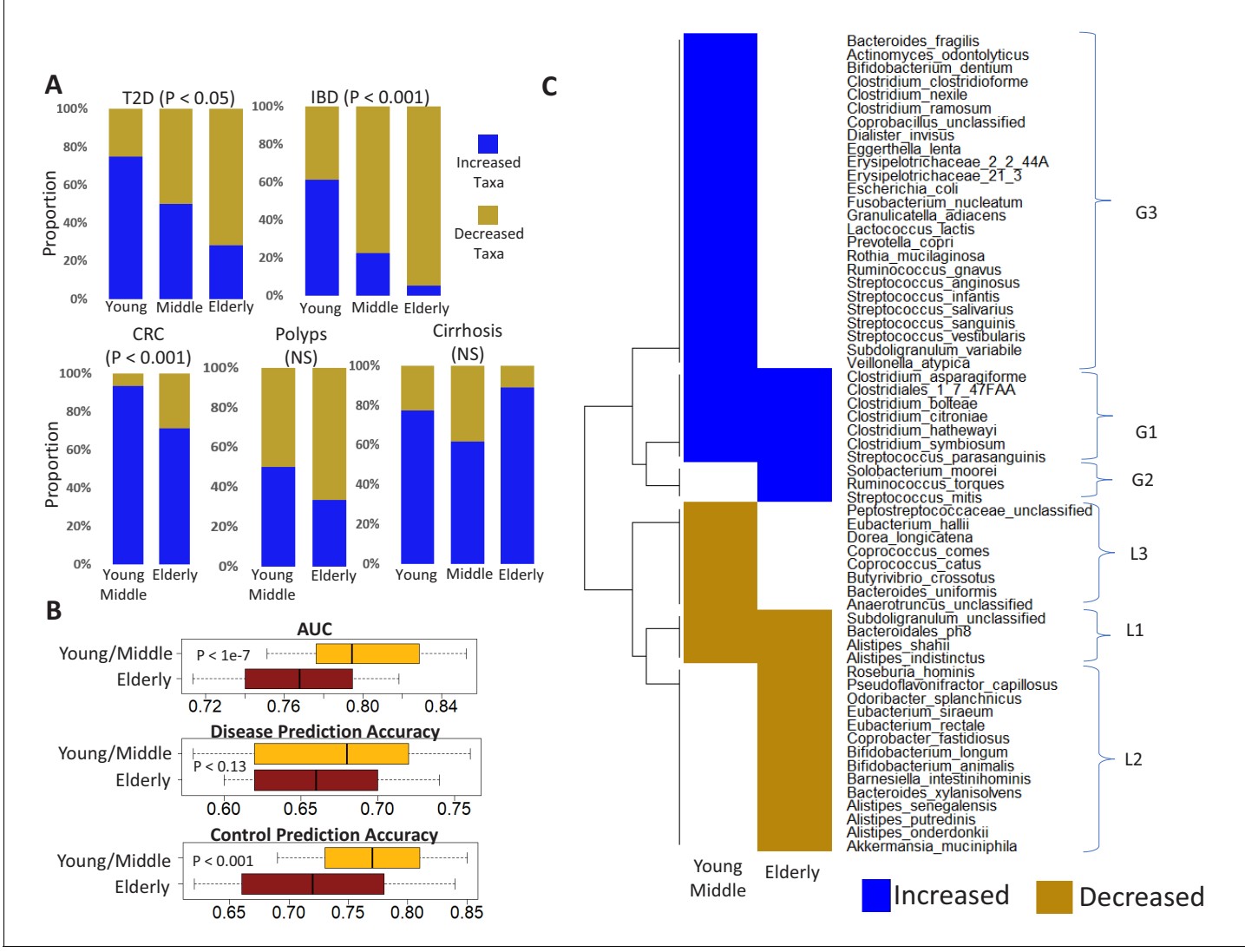

**Figure 5.** Age-related microbiome changes affect taxon abundance alterations for specific diseases, as well as the microbiome response shared by multiple diseases. (A) Comparison of the relative proportions of more abundant and less abundant disease-specific marker taxa across the young, middle-aged and elderly age-groups for the five diseases. For each disease-age-group scenario, we checked for the directionality (increased abundance in disease v/s decreased in disease) of association of the corresponding top disease-predictors by comparing their abundance trends in the control and diseased samples belonging to the specific age-groups (See Materials and methods). To ensure that the results thus obtained were not affected by regional variations in microbiome composition, we again restricted these comparisons to the disease-specific continent cohorts. (B) Comparison of the disease prediction AUCs, the disease classification sensitivity and control classification specificity of generic disease prediction models obtained for the elderly and young/middle-aged groups. Overall, the generic disease classifiers had a significant decrease in performance in the elderly age groups, indicating that shared microbiome response may be reduced in the elderly. Moreover, the loss of performance was especially significant with respect to the discrimination of control samples from disease (C) Heatmap of marker species showing consistent trends of either increase or decrease in at least two diseases in the elderly and young/middle-aged groups. Blue indicates consistent increase in two or more diseases, red indicates decrease in two or more diseases. Based on their patterns of increase or decrease across the two age-groups, the taxa could be classified into six groups, namely G1-G3 and L1-L3.

The online version of this article includes the following source data and figure supplement(s) for figure 5:

**Source data 1.** List of markers having significant increase (gain) or decrease (loss) of abundance with disease across the various age-groups.
**Figure supplement 1.** Comparison of the relative proportions of taxa increased and decreased in disease across the young, middle-aged and elderly age-groups for the five diseases.
**Figure supplement 2.** Comparison of beta diversity (measured as spearman distances) within the gut microbiome of controls from the young/middle and elderly age-groups from (A) Asia (B) Europe and (C) North America.

The overlap of altered taxa across diseases reported in one study was 51% (*Duvallet et al., 2017*), while another study reported that generic control *versus* disease classifiers trained by agglomerating disease samples from multiple studies could still distinguish controls from disease with an AUC of greater than 0.8 (*Pasolli et al., 2016*). Given that microbiome composition changes with age, we investigated the effect of age on the extent of these shared disease responses. We designed generic disease classifiers (using Random Forest), taking equally sized sub-samples of controls and diseased individuals (containing equal number of samples from each disease to prevent disease/age-group specific biases in classification performance) (See Materials and methods and *Figure 1—figure supplement 3*). While the performances of the generic disease prediction models in the young/middle-aged was high (median AUC: 0.79) and similar to those reported by earlier studies (*Pasolli et al., 2016*), the same models applied to data from elderly subjects had significantly lower performance AUC (p<1e-7) (*Figure 5B*). Moreover, in these models, while no significant differences were observed with respect to the disease prediction sensitivities (p<0.13), the specificity of prediction (that is the accuracy of identifying healthy individuals) was significantly lower for the elderly. This was not an effect of the differential representation of samples from the different diseases, as we had ensured equal representation of all diseases across all age-groups. Thus, in contrast to previous meta-analyses, the shared disease response was significantly lower across elderly subjects, primarily with respect to the discrimination of non-diseased individuals. Furthermore, our clarification of the effect of age on disease-associated taxa (*Figure 3—source data 1*; *Figure 3—source data 2*; *Figure 5—source data 1*) provides a refined set of features for improved microbiome-based diagnostics for these diseases. The above disease-independent changes in the pattern of microbial alterations as well as the inability of the generic disease classifiers to distinguish non-diseased controls were intriguing and could be a consequence of the loss of beneficial bacteria in the gut microbiome with ageing, which in turn could be driven by a multitude of factors like diet and medication. This increasing loss could lead to dysbiotic configurations characterized by higher inter individual variability, increased abundance of pathobionts (resulting in loss of disease signature) thereby making the microbiome more susceptible to diseases. Therefore, using intra age-group Spearman distances, we next checked whether the samples from elderly controls had significantly higher variability as compared to young/middle-aged controls, across the different continental regions (*Figure 5—figure supplement 2*). In line with our hypothesis, for both Europe and North America, the gut microbiome from elderly individuals was significantly more variable compared to young and middle-aged controls. However, this was not observed in the Asian (Chinese) cohort. Interestingly, the cirrhosis group neither shows an association with disease signatures nor the gain versus loss pattern contained only subjects from Asia.

Next, we sought to characterize the elements of the shared disease response. Notably, closer inspection of the directionality of the associations indicated specific taxa with consistent trends of association with multiple diseases (based on the trend shown in *Figure 5—source data 1*). The overall patterns of taxon gain or loss encompassed several trends observed by earlier studies (*Duvallet et al., 2017*; *Pasolli et al., 2016*). For example, *Streptococcus anginosus* and *Fusobacterium nucleatum* were detected as gained in multiple diseases. Similarly, species belonging to *Roseburia* spp. (*R. hominis*) were lost. We identified a total of 61 taxa that showed consistent directionality of association with multiple diseases in either young/middle-aged or the elderly age-groups (*Figure 5C*). Based on their differential detection profiles in the shared response across age-groups, we assigned these into six different groups, namely G1 (increased in disease across all age groups), G2 (increased in disease only in the elderly), G3 (increased only in young/middle-aged), L1 (decreased in disease across both), L2 (decreased only in the elderly) and L3 (decreased only in the young/middle-aged groups) (*Figure 5C*). Many of the species previously reported as associated with shared gain or loss across multiple diseases (*Duvallet et al., 2017*; *Pasolli et al., 2016*) belonged to the G3 group, that is, they showed similar trends of gain or loss in disease (as reported earlier) in the young and the middle-aged groups, but not in the elderly. These included the Streptococci, *Fusobacterium nucleatum*, and *Escherichia coli* and *Bacteroides fragilis*. In contrast, a separate group of species including *Ruminococcus torques*, *Solobacterium moorei* and *Streptococcus mitis* were associated with multiple diseases only in the elderly. Finally, we identified a distinct group of species (G1) that was gained across diseases in both elderly and young/middle aged groups. These included a group of Clostridia (*C. bolteae, C. symbiosum, C. hathewayi, C. citronae, C. asparagiforme*) (*Figure 5C*; see *Figure 5—source data 1* for individual diseases). These taxa have been identified in

separate studies of different diseases and/or disease-like states (T2D, Polyps, CRC and autism) (*Pequegnat et al., 2013*; *Qin et al., 2012*; *Sinha et al., 2019*; *Yu et al., 2017*), but are shown here, for the first time to be part of a shared gain response across diseases. Based on these findings, we hypothesize that this specific G1 group of taxa constitute a shared disease response associated with a general patho-physiological failure in the affected individual.

## Reproducible association of the G1 disease-positive markers with increased frailty in elderly individuals from the ELDERMET cohort

Frailty in the elderly is characterized by reduced function of multiple systems. We investigated if the taxon associations with frailty could be validated in the ELDERMET cohort (*Supplementary file 2*), for whom we had both shotgun metagenome and faecal metabolomic data. Using Random Forest regression (with five-fold cross validation), we could predict the frailty of an individual (testing both community- and residential care-dwelling subjects) based on the taxonomic composition of the gut microbiome (R = 0.79; *Figure 6A*; *Figure 6—figure supplement 1*). We then calculated the mean feature importance ranks in the frailty prediction model for the six gain/loss generic disease response groups (from *Figure 5C*). Validating our previous observations, the G1 group of taxa had the highest mean rank of feature importance scores for frailty (Functional Independence Measure (FIM)) prediction, indicating that this group had the highest predictive power for frailty in the ELDER-MET subjects (*Figure 6B*), followed by the G2 group (increased across multiple diseases only in the elderly). Validating our earlier findings on the elderly-specific markers of shared microbiome response, this was followed by the L1 group. An optimal number of eight taxa had the highest frailty predictive ability (*Figure 5C*; *Figure 6C*; *Figure 6—figure supplement 1*). All these taxa were more abundant with frailty (negatively correlated with FIM and positively associated with Frailty) (*Figure 6—source data 1A–B*). Five of the eight taxa belonged to the G1 group (*Figure 6C*; *Figure 5C*). Thus, these findings independently validated in the ELDERMET cohort our earlier identification of specific elderly-associated generic disease response groups (*Figure 5C*).

If microbiota alterations contribute to disease causation, microbial metabolites may be effector molecules. A distinguishing feature of the current study is that it is based on metagenomic data, as distinct from previous 16S-based meta-analyses of microbiome-disease (*Duvallet et al., 2017*). This allowed us to obtain taxonomic composition at finer taxa level, which in turn we could use to predict metabolite profiles of the microbiomes by exploiting experimentally validated functional profiles of the constituent taxa. For this purpose, we used the Virtual Metabolic Human database (*Noronha et al., 2018*), as well as a recent method to search experimentally-verified functional profiles representing the production and consumption patterns of different metabolites in various gut-associated species (*Sung et al., 2017*). We could thus collate more than 300 metabolic profiles (consumption/production/degradation) from 992 species (*Figure 6—source data 2*). A total of 82 metabolite profiles were observed to have significant association with FIM scores (Spearman Rho; FDR of less than 0.25) (*Figure 6—figure supplement 2*). We observed that this association analysis of metabolite profiles reflects frailty-associated changes with respect to bioavailability of specific compounds, many of which have been previously shown to have corresponding associations with health, thereby validating our findings (*Claesson et al., 2012*). Specifically, the onset of frailty is observed to be associated with an increase of SCFA consumption by gut-bacteria (accompanied by a concomitant decrease of the production of the SCFA butyrate), increased consumption of the beneficial amino acid Tryptophan as well as the increased production of the T2D-linked branched chain amino acid Threonine. We first focussed on the group of frailty marker taxa and identified 13 metabolic profiles (*Figure 6D*) that were significantly associated with the eight frailty-marker taxa (See Materials and methods). This subset of 13 metabolite profiles alone could predict frailty with a R value of 0.60 (between the actual and predicted FIM values), which is significantly higher than that obtained after removing these features from the metabolite profile (*Figure 6D*; Top panel second from left). The first set of key metabolic profiles included the degradation of primary bile acids, namely cholic acids (CA) and chenodeoxycholic acids (CDCA) to produce hydrophobic secondary bile acids (lithocholic acid: LA; deoxycholic acid: DCA). We validated this predicted metabolic functionality against the experimentally measured fecal metabolomic profiles for ELDERMET subjects, where the fecal metabolomes of frail individuals were characterized by significantly higher levels of LCA (along with its derivatives) and DCA (p<0.006) and significantly lower levels of CA and CDCA (p<0.02), as compared to the non-frail individuals (*Figure 6D*). The two species associated with this

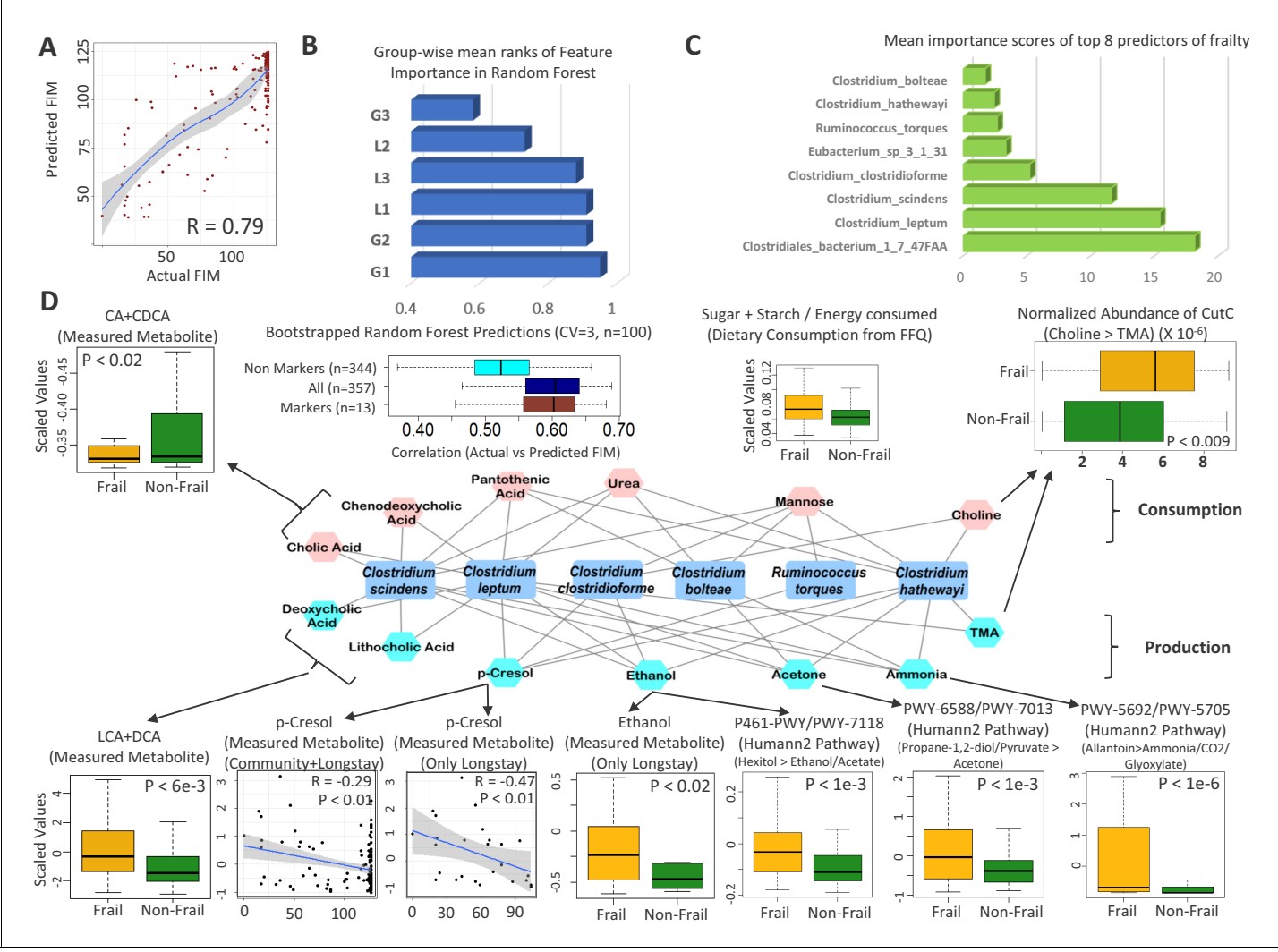

**Figure 6.** Frailty-associated markers have shared positive associations across multiple diseases in both age groups and have a specific metabolic signature. (A) Actual FIM values versus FIMs predicted by Random forest of microbiome features of the elderly individuals of the ELDERMET cohort living in Community or Residential care (Longstay). (B) Mean ranks of the various taxonomic groups (identified in *Figure 3*) for the prediction of FIM (an inverse measure of frailty) in the ELDERMET cohort. (C) Variable Importance Scores of the eight markers with the highest predictive power in the Random Forest models for prediction of FIM. A comparison of the abundance of markers between HighFIM and LowFIM individuals indicated that all of these markers were associated with frailty state. (D) The network in the central panel indicates the 13 metabolite profiles significantly associated with the top markers. Taxon markers are indicated in the center. Consumption profiles are in the upper half (in pink octagons) and Production profiles are on the lower panel (in yellow octagons). Edges indicate presence. Second from the left in top panel are the correlations between predicted and actual FIM values obtained for iterative bootstrapped Random Forest models (training on 20% and testing on the rest 80%) using only the 13 metabolite profile markers of (D), all metabolite profiles and all metabolite profiles removing the 13 metabolite markers. Top and bottom panels show the validation (indicated by arrows) obtained for the predicted metabolite markers using either the measured metabolites, dietary consumption profiles, specific microbial pathway abundances as well as the CutC gene family abundances identified using humann2 (shown either as boxplot comparing the profiles between Frail and Non-Frail individuals or scatterplots showing correlations between the measured metabolite level and the FIM value of the individuals). A total of 11 of the 13 metabolites could be validated using either of these strategies.

The online version of this article includes the following source data and figure supplement(s) for figure 6:

**Source data 1.** Top 17 predictive features for (A) FIM and (B) Barthel Score in the ELDERMET cohort.
**Source data 2.** Predicted metabolite map of species in this study based on combined pathway-taxon associations from *Noronha et al. (2018)* and *Sung et al. (2017)*.
**Figure supplement 1.** Frailty-prediction using Random Forest models and the identification of the topfrailty-predictive taxonomic features.
**Figure supplement 2.** Violin plots showing the Metabolite consumption and production profiles that were significantly associated with FIM scores (with Spearman Rho FDR < 0.25).

*Figure 6 continued on next page*

*Figure 6 continued*

**Figure supplement 3.** Heatmap based representation of the metabolic signatures associated with taxa gain/loss groups defined in main text *Figure 4C*: (A) G1-G3 (B) L1-L3.

functionality, namely *Clostridium scindens* and *Clostridium leptum*, did not belong to either the G1 or G2 groups of species (contrary to their frailty-association in this study). These species were previously implicated in *Clostridium difficile* resistance and IBD, respectively (*Buffie et al., 2015*; *Manichanh et al., 2006*). Contrary to their therapeutic relevance, this specific functionality of these species, namely production of higher levels of hydrophobic bile acids, has also been associated with the onset of CRC and non-alcoholic fatty liver disease (NAFLD) (*Tsuei et al., 2014*). One of the recent meta-analyses of CRC gut microbiomes also predicted increased production of secondary bile-acids to be a key CRC-specific functional signature (*Wirbel et al., 2019*).

Two other metabolite functionalities associated with the frailty-specific markers that were also validated in the fecal metabolome profile were p-cresol production (measured p-cresol negatively correlated with FIM: p<0.01; R = −0.29 with FIM across community and longstay; R = −0.47 for only long-stay) and ethanol production (significantly higher levels in the fecal metabolome of frail individuals in the long-stay cohort, as compared to the non-frail individuals: p<0.02). While p-cresol is a cytotoxic compound (produced by specific gut bacterial species including *C. difficile*) that is associated with microbiome alteration, large bowel cancer and insulin resistance (*Fau et al., 1976*; *Khan et al., 2014*), high level ethanol production in the gut has not only been linked to incidence of NAFLD, atherosclerosis and SIBO, but also to an increase in the levels of inflammatory markers (*Elshaghabee et al., 2016*). Similarly, production of acetone, ammonia ($NH_3$) have been associated with increased cytotoxicity, small intestinal bacterial overgrowth, as well as insulin resistance (*Baskaran et al., 1989*; *Ghoshal et al., 2017*; *Khan et al., 2014*). Although fecal metabolome data was not available for ExperimentHub microbiome samples, the abundance of specific microbial pathways (obtained using humann2) was associated with the production of these metabolites, namely propane-1,2-diol to acetone, and allantoin to Ammonia/CO2/Glyoxylate was predicted to be significantly higher in the microbiome of the frail individuals (*Figure 6D*; Lower panel). Another functionality associated with the frailty markers was choline consumption and Trimethylamine (TMA) production. Specific gut microbial species including *C. citroniae*, *C. clostridioforme* and *C. hathewayi* degrade choline to trimethylamine, which is liked with atherosclerosis (*Martínez-del Campo et al., 2015*). Profiling the abundance (normalized by sequencing depth) of the specific bacterial CutC enzyme catalysing this conversion revealed significantly higher representation in the gut microbiome of the frail individuals (*Figure 6D*; Upper right panel). This association with CRC has also been reported in one of the meta-analysis datasets used in the current study (*Thomas et al., 2019*). Thus, there is a *prima facie* mechanistic case whereby this group of microbiome markers could functionally drive the host to an increasingly pan-disease susceptible state, also accelerating the onset of frailty. Interestingly, identifying similar metabolite associations across the G1-G3 markers indicated overlapping metabolite production signatures of the frailty-associated markers with even the G3 groups of taxa (specifically the production of Ethanol and $NH_3$), indicating that the different taxa groups enriched in a generalized microbiome-disrupted state, irrespective of differences in their composition, contain certain metabolic signatures that may drive the intestinal eco-system to a state detrimental to the host (*Figure 6—figure supplement 3*). Similarly, comparing the metabolic signatures of the taxa in the G1-G3 groups with those of the different groups of lost taxa L1-L3, also revealed an interesting pattern whereby the 'gained' groups were disproportionately associated with the consumption of simple sugars (lactose, psicose, xylose, sucrose, raffinose). The 'lost' groups in contrast were enriched for consumption of dietary prebiotics like xylo-oligosaccharides, fructo-oligosaccharides, inulin and production of butyrate and succinate. A recent study of the gut microbiome of Thai individuals migrating to the US observed the loss of metabolic functions linked to degradation of dietary fibers like xylan, arabinoxylan, cellobiose, pullulan, glucomannan and resistant starch, with increasing duration of stay leading to a broader loss of microbiome function (*Vangay et al., 2018*). These patterns, reflective of the metabolic capabilities of specific microbial groups, are especially important for diet-based microbiome restoration strategies in the elderly, where loss of control-associated taxa has a stronger effect on disease, as compared to the young and the middle-aged.

## Discussion

By identifying specific age-linked microbiome associations for different diseases, this study can potentially inform the development of microbiome-based diagnostic strategies customized for specific age-groups. Accounting for age in disease-microbiome classifiers has clarified the microbiome alterations common to many diseases and shortened the lists of taxa specifically implicated in common non-communicable diseases. Furthermore, distinct changes observed in the overall directionality of taxon associations (e.g. the gradual shifting to a loss-of-taxa phenotype in the elderly) highlight that microbiome restoration strategies may be more crucial for disease amelioration in elderly subjects (as opposed to antibiotic-based eradication regimens). This is especially important since specific changes are observed with ageing that can potentially make the host more susceptible to certain microbiome-related diseases. The above pattern was further observed in our analysis using generic disease classifiers. The specific loss of the ability of the classifiers to distinguish between cases and controls further indicates that with ageing, there is loss of health-associated microbiome signatures. Interestingly, these findings resonate with the results of one of our earlier studies, where with increasing age, a decrease of the core microbiome accompanied by an increase of pathobionts was observed (O'Toole and Jeffery). Finally, reproducible identification of a specific set of species markers associated with multiple diseases and their specific metabolite profile (sugar consumption and the production of a range of metabolites that are detrimental for the host) indicates that the acquisition of this subset of disease-associated taxa can shift the metabolic state to a disease-like state.

Separating correlation from causation in microbiome-disease studies requires identification of putative taxa correlating robustly with disease, which will be improved by accounting for age in microbiome-disease association studies. Many of the features in the current analysis were not reported in the original publications. There could be several reasons for this, namely variation in the comparative analysis protocol used in the original studies as compared to those in the current study; more power and robust signatures in the current analysis by virtue of combining multiple studies; study-wise biases in the number of samples belonging to different age-groups. In this regard, a key aspect is the relative stability of the age-specific differential associations across regions/ethnicities. In this study, we have used a combination of Random Forest, linear models as well as validations at different levels of regional homogeneity to not only reduce the effects of regional variations on microbiome signatures but deconvolute the effect of general ageing on these age-specific disease signatures. However, biases in number of samples for certain regions in specific disease-age-group scenario still exist. Currently, a majority of datasets in the curatedMetagenomic data repository and the other validation cohorts are from North America, Europe and China. However, future availability of microbiome data from disease cohorts especially from Non-Western populations will further clarify the age-specific trends of disease classification. This is especially important, since in the current study, we identify region-specific changes across age-groups, where in, while the gut microbiomes of elderly controls from North America and Europe have significantly higher variation as compared to region-matched young/middle-aged controls, no such differences are observed within the Asia (Chinese) populations. Future availability of new disease-specific shotgun microbiome datasets will further enable cross-cohort validations of disease signatures (as was performed in the current study for CRC) on other diseases. Another factor that needs to be considered is the effect of diet. Many of the taxa that are depleted in disease are also taxa whose growth is promoted by complex carbohydrate consumption. Different populations have specific dietary patterns and these associations need to be factored into suitably designed future prospective studies.

We could not control for medication intake in subjects/data from the ExperimentHub, because this data was not available (except for antibiotic treated subjects that were removed), and which can be a major confounder (*Forslund et al., 2015*). Previous studies have reported the association of some of the disease markers with medication intake. For example, Streptococcaceae have been shown to be associated with intake of Proton Pump Inhibitors (*Jackson et al., 2018*). Another recent study investigating the effect of metformin on the gut microbiome of two of the T2D cohorts analysed in the current study identified *Lactobacillus salivarius* to be affected by metformin, accompanied by an increase in *Escherichia* (*Forslund et al., 2015*). However, none of the other G1 and G2 markers were reported to be affected. The FranzosaEA_2018 dataset (added along with the curatedMetagenomicData repository) contained drug usage information for Immuno-suppressants,

Steroids and Mesalamine usage (*Franzosa et al., 2019*). For these three drugs, we checked whether the abundance of shared disease markers was affected by medication intake (*Supplementary file 3*). We could only identify associations for Peptostreptococcaceae (decreased in mesalamine treatment), *Coprococcus comes* (increased in mesalamine treatment) and *Barnesiella intestinihominis* (decreased in steroid treatment). Moreover, adjusting for medication did not alter abundance of taxa predictive of frailty in the ELDERMET cohort (for which the medication usage data was available). Each of the top frailty-associated markers were not only observed to have significant associations with the different frailty measures even after taking into account for the effect of the selected sets of medication type, but also identified in the top 15 predictors of frailty, independently in the individuals with high or low medication (*Figure 5—source data 1c*). However, the lack of associations with this limited list of medications does not completely rule out the possibility of medications being partly responsible as a driver for these differential associations. Consequently, for diseases like T2D, the results obtained here have to be treated with caution. Future cohort studies should record all measurements of diet and medication intake, so that the complex interactions of age, lifestyle, diet, microbiome and health may be further elucidated. Another confounding factor may be age of disease presentation. For example, a recent study identified cancer stage-microbiome interactions (*Yachida et al., 2019*); however, we also note from the published data that 51% of the younger subjects (less than 60 years of age) presented with Stage III/IV cancer, whereas a significantly higher percentage (71%) of older subjects presented with Stage I/II cancer (p<0.01). Variation in disease presentation in different age groups could also influence the disease-microbiome signatures, and therefore needs to be considered in future studies.

The last concern is methodological and pertains to the non-independence between the training samples in the iterative RF classifiers generated for each disease-age-group scenario. Classifiers generated in each iteration are likely to share overlap of training (and/or testing) samples, thereby potentially resulting in narrower AUC distribution which could inflate significance values. However, the magnitude of the differences as well as consistency of the results (using multiple methods and study cohorts) clearly indicate the reliability of the results obtained in this study.

# Materials and methods

## Key resources table

| Reagent type (species) or resource | Designation | Source or reference | Identifiers | Additional information |
| --- | --- | --- | --- | --- |
| Software/Algorithm | curatedMetagenomic-Data | *Pasolli et al., 2017* Available as a R-library | | Version as in Oct, 2018 |
| Software/Algorithm | R packages: 1. randomForest 2. pROC 3. lmtest 4. ade4 5. vegan 6. dunn.test | Availabe as R packages from CRAN. | | Latest versions as on Oct, 2018 |

## Data collection from the curatedMetagenomicData repository

Since the focus of investigation was the gut microbiome, we first selected a subset of the curatedMetagenomicData, containing 4195 stool samples (annotated as 'body_site': stool) (*Pasolli et al., 2017*). We subsequently removed the samples which have not defined age and study-condition, thereby filtering the dataset to 3580 samples. From this set, we removed samples having age less than 20 years of age (retaining 2564 samples). Notwithstanding the uniform bioinformatics analysis strategy applied to this data, two major factors that may contribute an artefactual bias in multi-cohort microbiome datasets (and which were available in the metadata) are the read-length (obtained from the sequencer) and DNA extraction methodologies (which are study-specific). To test the effect of these factors, we first removed the samples from Peruvian, African and Fijian individuals in order to remove the confounding effects of region/life-style-specific), along with those from hospitalized individuals. Subsequently, on the remaining subset, we evaluated the effect of these factors using envfit on the species level profiles by first visually comparing the differences using PCoA and

then testing the confidence of these differences using envfit (https://cran.r-project.org/web/packages/vegan/vegan.pdf). For this purpose, we performed 20 boot-strapped envfit iterations, each time taking a subset of samples (sub-sample size: 200) and computing the $R^2$ and the significance (P-value) of the differences, and then comparing the distribution obtained with that obtained using a null distribution obtained by taking 200 sub-samples (after permuting the labels). We established that while read lengths had a marginal effect (R = 9e-3, p<0.09) (*Figure 1—figure supplement 1A*), samples from one of the studies (SchirmerC_2016) (*Schirmer et al., 2016*), using a DNA extraction technique tagged as 'Illuminakit' in the metadata, had a distinct taxonomic profile. We removed the 465 samples of this study from all further analyses, thereby reducing the effect of extraction methodology on taxonomic profiles (p<0.06; *Figure 1—figure supplement 1B–C*). To this compiled list, we added the samples from four recently published datasets (one IBD-specific dataset of 220 samples, referred to as 'FranzosaEA_2018' [*Franzosa et al., 2019*]; three CRC-Specific datasets referred to as 'WirbelJ_2019', 'ThomasAJ_Cohort1' and ThomasAJ_Cohort2 ' [*Thomas et al., 2019*; *Wirbel et al., 2019*]). This repository, along with the 189 shotgun sequenced samples from the ELDERMET cohort, resulted in a total of 2564 samples. Schematic representation of the workflow used for preparing a core set of 2564 gut metagenomic samples derived from the publicly available datasets (curatedMetagenomicData and the four newly added cohorts) and the ELDERMET repository is provided in *Figure 1—figure supplement 2*. The details of the samples belonging to the curatedMetagenomicData and FranzosaEA_2018 dataset included in the current study are provided in *Supplementary file 1*. The clinical metadata of the ELDERMET samples are listed in *Supplementary file 2*. In this analysis, the taxonomic composition obtained using the metaphlan2 pipeline was obtained at the microbial species level. We have used the term 'species' and 'taxa' in this manuscript to refer to a taxonomic unit below the level of genus.

## Effect of host-associated factors, including age groups on the microbiome profile

We specifically investigated the metadata of the samples listed in *Supplementary file 1*. Some of the metadata were observed to be redundant and had similar associations (examples included groups like Country and Dataset Name; Age and Age-category; Study condition and Disease; Antibiotics current use and Antibiotics family). In these cases, we retained the former metadata and removed the latter ones. We added another region-specific metadata, namely Continent, for reasons explained in the subsequent section. Subsequently, we filtered out those metadata present in less than 30% of the samples. A total of six metadata remained. We obtained the association of each of these metadata using PERMANOVA (using the adonis function of the vegan R package). Given that DNA extraction methodologies still had a marginal effect on the gut microbiome composition ($R^2 = 0.019$), the PERMANOVA analyses for each of the metadata were performed after adjusting for the DNA extraction method as a confounder. The PERMANOVA analysis (for each metadata) was performed using the adonis function of the vegan package using the following pseudocode formula: adonis(species ~ dna_extraction_method + metadata).

For investigating the variation of the microbiome with age across the adult-hood landscape, the individuals were binned into three age groups namely young (20–39 years of age), middle-aged (40–59 years) and elderly (60 years and above). We removed the antibiotic-treated subjects from all subsequent analyses. Principal component analysis of the microbiome profiles of the samples belonging to the three age-groups was performed and plotted using dudi.pco and s.class function of the ade4 R package. The significance of the association was obtained using the PERMANOVA (adonis function) implemented in the 'vegan' R package (with 'country' and the DNA extraction method as a confounder).

## Grouping samples into country-/continent-specific bins

Given that regional factors had the highest effect on the microbiome composition, it was important to ensure regional homogeneity for comparative disease-association analysis. However, the majority of disease-specific cohorts either displayed significant differences in age difference in the age of the control and diseased individuals (i.e. they were not age matched) or biases for disease patients from specific age-groups (*Figure 2*). For each disease, collating samples from the same country or continent-level as the disease cohorts would bypass the issue of limited sample numbers across the age-

groups, whilst maintaining regional homogeneity of the cohorts (*Figure 1—figure supplement 3A–B*). To compare the overall effects of the two regional factors, country and continent, on the microbiome profiles we performed bootstrapped PERMANOVAs (by taking 20% subsets) within the control individuals. The results indicated that, although continent was observed to have a marginally lower effect on the microbiome composition compared to nationality (country), performing repeated bootstrapped comparisons indicated the effect of continent to have a higher significance (calculated as -log of Adonis P-values) than the country on the microbiome profiles (*Figure 1—figure supplement 3C*). For each disease, the country/continent specific affiliations of the disease cohorts were first obtained. Subsequently, we performed all the investigations pertaining to each disease by pooling samples belonging to the same country as the corresponding disease cohorts (referring to them as disease-specific country-level bins) (*Figure 1—figure supplement 3D*). This was expected to optimally homogenize the region-specific variations, while ensuring enough representation of various diseases (and controls) across age-groups. Wherever applicable, we have adopted a similar disease-specific regional grouping strategy at the level of continents.

## Investigating the interaction between disease signatures and age-group

For each disease, we performed PERMANOVA within the corresponding disease-specific country bins investigating the effect of the interaction of Disease and Age-group (Disease:Age-group) after adjusting for the effects of Country, and the independent effects of Disease and Age-group. Briefly, the pseudocode of the formula is provided below: adonis(species ~ country + disease + age-group + disease:age-group).

## Disease classification using Random Forest (RF) Models

If diseases have age-group specific signatures, then classifiers trained on the same age-group would have significantly better performance when tested on the same age-group as compared to that when tested on different age-groups. To evaluate the performance of disease classifiers trained on one age-group on disease prediction in either the same (Same Age-group classification) or different age-groups (Different Age-group classification), we adopted the following strategy (*Figure 2—figure supplement 1*). For each disease-age group combination, we performed 100 iterations, such that in each iteration, we trained the classifier on a subset of disease and the same number of control samples (50% of the minimum number of diseased samples across any age-group; denoted as 'training subset', the disease-specific training subset sizes as defined in *Figure 2—source data 1*). The evaluation of each of these disease classifiers for 'Same Age-group' and 'Different Age-group' classification was then performed using two approaches as mentioned below.

In the first approach, we created two 're-sampled' test sets. While one contained Y diseased and Y control individuals from the same age-group (but not included in the training sub-set) (Same Age-group test set), the other contained Y diseased and Y control individuals from the other two age-groups (Different Age-group test set) (with Y defined as in *Figure 2—source data 1*). We tested the classifier on each test set and computed the AUCs. Testing the same classifier on both test sets ensured that the observed variations in disease prediction performance was not due to differences in the subject sub-samples used to create the classification models. Further to ensure that we don't have biases introduced because of the selection of test sets (same age-group, different age-group), we repeat above steps 20 times (per classifier in each iteration) and computed the median AUCs for both the Same Age-group classification and the Different age-group classification. These median AUCs obtained for the Same Age-group and Different age-group classification for each of the 100 iterative sub-sampled classifiers was compared using Wilcoxon Signed Rank tests (to check if the performance of the classifiers significantly varied when tested on the same or different age-groups).

In the second approach, using Permutation test framework, we tested whether observed difference in classification AUCs (*i.e* AUC for Same Age-group classification – AUC for Different Age-group classification) was significantly different than what would be expected by random. For this, we needed null distribution of empirical differences of AUC for the same sub-sampled classifier. For this purpose, for each of the 20 iterations corresponding to each of the 100 sub-sample based RF classifier models (as described in the previous paragraph), we first merged the two 're-sampled' test sets (Same Age-group test set and Different Age-group test set), then permuted the age-group labels of the subjects, creating two 'Permuted' test tests (Permuted test set 1 and 2), tested the classifier

model for each of the Permuted test sets and finally computed the AUC differences obtained for the same classifier between the two permuted test sets. For each of the RF classifiers (across the 100 iterations), we computed the medians of the differences of AUCs (for the two permuted test sets) across the 20 iterations. The median difference of the AUCs obtained for the actual test (for Same Age-group – Different Age-group) and the permuted tests (Permuted Test set 1 – Permuted test set 2) obtained for the 100 iterations are then compared with Wilcoxon signed rank-tests. The objective of these permutation tests was to reduce the effects of correlated errors associated with certain 'aberrant' samples in influencing the disease prediction performance of RF classifiers for specific age-groups. This permutation test is expected at least ensure that the correlations introduced by sampling and by testing the same classifier on multiple cohorts will also be present in the null distributions. For both the approaches, the p-values of comparison across age-groups were corrected using Holms' correction.

For a given disease, to ensure that the observed changes were not artefactual consequences of differences in sizes of training and testing subsets, we kept the training and testing subset sizes constant across all training age-groups. The classification AUCs, Specificities and Sensitivities were computed using the various modules in the pROC package.

## Identifying the top disease-associated marker taxa for the different age-groups

For each disease-age group scenario (as described above), we ranked the species-level taxa in decreasing order of their mean importance scores (mean decrease in GINI) across all 100 iterations. The iterative sub-sampling based Random Forest analysis avoided overfitting and, by virtue of the uniformity of training and test sizes for each disease across all age-groups, ensured that the differences across age-groups were not a simple consequence of differences in the number of diseased and control individuals. The objective of the iterative approach was also to identify taxa that showed stable association with disease irrespective of the sub-sampling population. This aimed to identify taxa that were always considered for the classification model irrespective of the sampled population of diseased and controls (Mean Decrease of GINI of at least 0 across at least 95% of iterations). So, for each disease/age-group scenario, we aimed to identify that minimum percentile threshold above which the taxa were considered for the classification model across at least 95% of the iterations (*Figure 3—figure supplement 1A*). We observed that for taxa, having a percentile score of at least 85, included in training models across at least 95% of the iterations across all the 13 disease age-group scenarios. Further plotting the mean feature scores of taxa arranged in terms of their percentile scores, we observed that the mean feature importance scores remained stable and low till the 80-percentile score and started increasing considerably only after that (*Figure 3—figure supplement 1B*). Based on these two results, the taxa having importance scores in the top 15 percentile (that is higher than 85 percentile) were identified as the top 85 percentile predictors/markers for a given disease in that age-group.

## Validation of the shortlisted disease-associated taxonomic markers using linear models and identifying overlaps with taxa reported in original studies

In this linear model based validation strategy, within the disease-specific country-level bins, we modelled the (log transformed to the base 10) abundance of each species first as a function of disease-status and country and then as a function of the interaction of disease-status and age-group and country as an independent predictor as described below:

Model 1:
Log(Species)~Country + Disease + Age-group
Model 2:
Log(Species)~Country + Disease * Age-group
(which is equivalent to Log(Species)~Country + Disease + Age-group + Disease:Age-group)

The goodness of fit of both models were quantified using adjusted R-squares and AIC values, and the significance of improvement of Model two with respect to Model 1, was judged using log-likelihood tests. This took care of the country as a confounder as well as provided the extent to which the interaction of disease and age-group had an influence on the abundance of the species as

compared to the individual factors. Those species showing significant improvement in performance of Model two with respect to Model 1 (log likelihood ratio test p<0.05 one-sided), could be identified as ones for which age-group had significantly high influence on the disease associations. For the case of T2D, however, there were skews in the representation of the diseased samples across countries, where in the diseased samples in the young and middle aged group were only from the Chinese cohort, while those from the Elderly individuals had representation from both the Chinese and Swedish cohort. Consequently, to ensure regional homogeneity, in the above validation for T2D, we included young/middle-aged controls from only the Chinese cohort, while the elderly controls included those from both the Chinese and the European (Swedish for country-specific comparisons) cohorts. Taxa having significant differences in their RF feature importance scores (FDR < 0.01 at least one pair of age-groups using Dunns' test for IBD, T2D and Cirrhosis and Mann-Whitney Tests for CRC and Polyps), which were also validated as having log likelihood ratio test p<0.05 in the linear model based validation were identified as the final validated list of age-group specific markers. This was done because the validation of each taxa was independent and additional stringent thresholds during validation would result in loss of sensitivity.

In order to evaluate the prevalence of these age-linked markers in the known disease-microbiome associations, for four of the five diseases, we obtained the list of associated species (detected as either significantly different or as a marker with high disease predictive score) from the original published studies corresponding to each of the disease cohorts (*Feng et al., 2015*; *Franzosa et al., 2019*; *Karlsson et al., 2013*; *Qin et al., 2012*; *Qin et al., 2014*; *Vogtmann et al., 2016*; *Zeller et al., 2014*) (*Figure 3—source data 3*). For CRC, we utilized the list of markers provided in *Thomas et al. (2019)*. We specifically utilized the list from this study, as it had already performed a multi-cohort meta-analyses of CRC-specific taxa adopting a similar metaphlan2-humann2 based classification scheme. The list (as created above) was then compared with list of age-linked markers specific to each disease (*Figure 3—source data 3*).

## Reproducing the results of CRC-specific trends on the validation cohorts

Iterative age-group specific disease classifiers were devised and tested as described in *Figure 2—figure supplement 1* (by specifically setting the 'Training' and 'Testing' cohorts). For classifiers designed on the validation cohort, the feature score importance scores of the top 85 percentile markers (obtained during the iterations) were compared using Mann-Whitney U-tests between each age-group (in each cohort). For testing the reproducibility of the associations, the directionality of the 19 known markers and the significance of their differences in the two training cohorts were then obtained and compared. For investigating the stability of the microbiome signatures (obtained for the different age-groups) across training cohorts, we profiled the mutual variations in the feature rank profiles obtained in cohort with respect to those obtained for the other. For a given feature rank profile (obtained in a given cohort for a specific age-group), its mean Spearman distance to all the feature rank profiles obtained for the same age-group in the other cohort was obtained. This was then collated for all feature profiles for the same age-group in both the cohorts. The distribution of the distances would indicate the stability of the microbiome signature across cohorts. Lower cross-cohort distances would indicate stable reproducible signatures and higher cross-cohort distances signify a relative lack of signature.

The ageing associated changes in the prevalence of some of the CRC markers and their effect relative variation within disease and controls were profiled in the following manner. In the first step, we compared the prevalence rates of these markers in the elderly controls and the young-middle aged controls using Fishers' Exact tests separately for the two cohorts. The p-values obtained for each cohort were then merged using the Fisher method. The effect size difference (disease v/s control) distribution for the markers computed using age-group specific iterations wherein each iteration, we compared 20 disease samples with 20 controls and computed the effect size using Cohens' D. Empirically, a Cohens' D values of less than 0.2 indicates low effect and greater than 0.2 indicates medium to high effect.

## Determining the directionality of disease association in each disease-age group scenario

To obtain the directionality (increased or decreased in disease) of the of the microbiome changes, Mann-Whitney U tests were first performed for all taxa to compare their abundances in 'study-matched' control and diseased samples (for ensuring regional homogeneity) from the specific age-group (ref to *Figure 1C* for the Study cohorts). The taxa having significant differences (Mann Whitney Test p<0.05) in their abundances between control and diseased samples were identified along with their directionality (ie. increased in disease or decreased in disease). These analyses were finally re-performed at the level of continent (that is between 'continent-matched' disease and control samples) to ensure statistical power to detect the gain versus loss patterns. For each disease age-group scenario, the top 85 percentile markers having significant change in their abundance with FDR corrected p-values<0.01 (for the continent level comparisons) were then filtered (*Figure 5—source data 1*). The directionality of these markers was assigned based on the trends of their abundance patterns, as either 'Increased' or 'Decreased' in disease.

## Creation of generic disease prediction models and identification of shared disease markers

The generic disease prediction classifiers were developed in a manner similar to that shown in *Figure 1—figure supplement 3*. The only difference was the agglomeration of equal number (n = 10) samples from each of the diseases (rather those of a specific disease) as described below. For each age-group (young/middle-aged or elderly), a generic disease cohort was created by taking equally sized sub-samples from each disease (to remove biases in the classifiers originating from specific diseases). The sub-sample sizes were also kept the same across the age-groups to ensure uniformity in the testing and training sizes of the classifiers across all age-groups. The iteration was subsequently repeated five times using a different (but equally sized) subset of diseased and control samples (as described above). The AUC and sensitivities for the five repetitions were then merged and compared across the young/middle-aged and elderly. To remove regional biases in microbiome compositions affecting these results, all analyses were restricted within the disease-specific continent cohorts (*Figure 1—figure supplement 3D*).

## Identification of Frailty-associated markers

We used a random forest model to regress both the Functional Independence Measure (FIM) and the Barthel Score (both an inverse measure of frailty) of an individual from the microbiome profile. Random forest regression training was performed on 20% of the samples and tested on the remaining 80%. The ranked feature importance scores of the different species were then obtained. Microbiome features (that is the species) were ranked in decreasing order of their feature importance scores (mean decrease in GINI coefficient upon excluding the feature). The mean feature ranks for the different groups of species (G1-L3) were then calculated.

To identify the most predictive marker set, the regression was repeated by iteratively reducing the number of the top microbiome features, and the mean error was calculated for each iteration. For both FIM and Barthel Score, the number of top microbiome features for which the error was minimum was taken as the set of the top frailty-associated markers. For both FIM and Barthel Score, the minimum number of features were 8 and 6, respectively.

The median FIM values were computed for both the residential care cohort and the overall cohort (community + residential care). For each cohort, individuals having FIM values below and above the corresponding median were classified as 'Frail' and 'Non-Frail', respectively. To validate the metabolite signatures, measured levels of actual metabolites, abundances specific microbial pathways and gene-families (obtained using humann2) were then either compared between these groups of 'Frail' and 'Non-Frail' individuals or correlated with the FIM values.

## Creating metabolite species maps and obtaining the frailty-associated metabolic signature of a given group of species

We utilized the literature-curated experimentally annotated species to metabolite (production/consumption) associations available as part of the Virtual Metabolic Human database as well as those obtained in a recent meta-analysis by *Noronha et al. (2018)*; *Sung et al. (2017)*, to create a

species-to-metabolite map of more than 300 metabolite production and consumption profile corresponding to 992 species in a 0 (absent) and 1 (present) notation (*Figure 6—source data 2*). For each microbiome, the metabolite production/consumption capability was then obtained as the matrix inner product of the abundance profile of the species and the species-to-metabolite map thus obtained.

Next, we identified the frailty-linked metabolites associated with the eight taxonomic markers of frailty using a two-step strategy. First, we performed a correlation analysis of each metabolite profile (i.e the cumulated abundance of taxa previously associated in literature with a given metabolic capability as obtained above) with FIM scores and identified metabolite profiles that showed significant association with FIM scores (Spearman Rho; FDR of less than 0.25). Next, we identified which of these identified metabolite profiles were detected in the taxonomic markers of frailty (based on previous literature) at a rate significantly higher than the background detection (using our Fishers' exact test approach with FDR corrected $p < 0.25$.

## Data and code availability statement

The detailed description of the codes is provided in the methods section. The key in-house source codes used in this meta-analysis have been provided as *Supplementary file 4*. The shotgun data of the ELDERMET is available for download from the ELDERMET website at http://eldermet.ucc.ie/temp1/eldermet_shotgun_data_filtered_all_sample.tar. The shotgun data for the ELDERMET has also been uploaded at the European Nucleotide Archive (ENA) with the project accession number PRJEB37017.

## Acknowledgements

The authors are funded in part by Science Foundation Ireland (APC/SFI/12/RC/2273) in the form of a research centre, APC Microbiome Ireland. IBJ was supported by a Science Foundation Ireland grant (13/SIRG/2128).

## Additional information

### Funding

| Funder | Grant reference number | Author |
| --- | --- | --- |
| Science Foundation Ireland | APC/SFI/12/RC/2273 | Tarini S Ghosh<br>Paul O'Toole |
| Science Foundation Ireland | 13/SIRG/2128 | Ian B Jeffery |

The funders had no role in study design, data collection and interpretation, or the decision to submit the work for publication.

### Author contributions

Tarini S Ghosh, Conceptualization, Data curation, Formal analysis, Validation, Visualization, Methodology; Mrinmoy Das, Resources, Data curation; Ian B Jeffery, Conceptualization, Formal analysis, Supervision, Validation, Visualization; Paul W O'Toole, Conceptualization, Supervision, Visualization, Project administration

### Author ORCIDs

Tarini S Ghosh https://orcid.org/0000-0001-9570-0365
Paul W O'Toole https://orcid.org/0000-0001-5377-0824

### Decision letter and Author response

Decision letter https://doi.org/10.7554/eLife.50240.sa1
Author response https://doi.org/10.7554/eLife.50240.sa2

# Additional files

## Supplementary files

- Supplementary file 1. Details of the samples in (A) the curatedMetagenomicData repository, FranzosaEA_2018 (*Franzosa et al., 2018*) dataset and (B) WirbelJ_2019 (*Wirbel et al., 2019*) and ThomasAJ_Cohort1 and ThomasAJ_Cohort2 (*Thomas et al., 2019*), used in the current study.

- Supplementary file 2. (A) Clinical Metadata of the ELDERMET Subjects (Code for Stratification: 1 = Community; 2 = DayHospital; 3 = Rehab; 4 = Longstay) and (B) Taxa abundance of each subject obtained using Metaphlan2.

- Supplementary file 3. Comparison of the abundances of the G1-G3 and L1-L3 markers in patients (of the FranzosaEA_2018 cohort) with and without different medication intakes as: (A) For patients with and without Mesalamine (B) For patients with and without Immunosuppressants (C) For patients with and without Steroids.

- Supplementary file 4. Codes and RData files for the key meta-analyses performed in this study, along with the corresponding Readme file.

- Transparent reporting form

## Data availability

The present study is a meta-analysis of previously published cohorts. The details of the datasets used in the current study have been uploaded as Supplementary Files 1 and 2. The datasets were obtained from the curatedMetagenomicData repository and the ELDERMET cohort. Source codes for the key analyses protocols have been provided as Supplementary File 4. The shotgun data of the ELDERMET is available for download from the ELDERMET website at http://eldermet.ucc.ie/temp1/eldermet_shotgun_data_filtered_all_sample.tar. The shotgun data for the ELDERMET has also been uploaded at the European Nucleotide Archive (ENA) with the project accession number PRJEB37017.

The following dataset was generated:

| Author(s) | Year | Dataset title | Dataset URL | Database and Identifier |
|---|---|---|---|---|
| Ghosh TS | 2020 | Identifying frailty-associated markers in Elderly Irish Individuals | https://www.ebi.ac.uk/ena/data/search?query=PRJEB37017 | EBI European Nucleotide Archive, PRJEB37017 |

The following previously published datasets were used:

| Author(s) | Year | Dataset title | Dataset URL | Database and Identifier |
|---|---|---|---|---|
| Franzosa EA, Sirota-Madi A, Avila-Pacheco J, Fornelos N, Haiser HJ, Reinker S, Vatanen T, Hall AB, Mallick H, McIver LJ, Sauk JS, Wilson RG, Stevens BW, Scott JM, Pierce K, Deik AA, Bullock K, Imhann F, Porter JA, Zhernakova A, Fu J, Weersma RK, Wijmenga C, Clish CB, Vlamakis H, Huttenhower C, Xavier RJ | 2019 | Gut microbiome structure and metabolic activity in inflammatory bowel disease | https://www.ncbi.nlm.nih.gov//bioproject/400072 | NCBI BioProject, PRJNA400072 |

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
