## [Decision Letter]

**Acceptance summary:**

This work emphasizes the importance of controlling for age in microbiome research and provides strategies for adjusting for this confounding factor across multiple disease areas. These results also set the stage for follow-on studies of the mechanisms responsible for these associations and their relevance to health and disease.

**Decision letter after peer review:**

Thank you for submitting your article "Adjusting for age improves identification of gut microbiome alterations in multiple diseases" for consideration by *eLife*. Your article has been reviewed by three peer reviewers, including Peter Turnbaugh as the Reviewing Editor and Reviewer #2, and the evaluation has been overseen by Wendy Garrett as the Senior Editor. The following individuals involved in review of your submission have agreed to reveal their identity: Catherine A Lozupone (Reviewer #1).

The reviewers have discussed the reviews with one another and the Reviewing Editor has drafted this decision to help you prepare a revised submission.

Summary:

In this paper, the authors perform a meta-analysis of publicly available shotgun metagenomics datasets and identify interactions between age and microbiome associates with different diseases. Overall, this work addresses an important question. In addition to yielding insights into microbiome correlates across a variety of diseases when controlling for age in this particular meta-analysis, it provides guidance to other researchers that age is an important component to consider in study design when conducting microbiome/disease research. The manuscript is well written and state-of-the-art analysis techniques are applied. The overall conclusions are interesting in that random forest classifiers in the elderly group do not do a good job classifying the young (and vice versa), and that there are shared microbiome signatures with diseases that are affected by age.

Essential revisions:

1) The evaluation relies on random forests, a tool for classification, when the paper is really more about adjusting for confounders, which RFs don't intrinsically handle. The authors might be better served using an established meta-analytic framework like a mixed effects model. In order to use the RF for their purposes, the authors have to subsample their data extensively and have to perform a lot of pairwise analyses where controls from the same continent are pooled (and differences between countries on the same continent ignored). A hierarchical model should be able to capture and adjust for effects of continent and country in a way that made more efficient use of the available samples, would offer some protection against unbalanced examples, and would have more easily interpretable coefficients. While it's true that random forests have better classifier performance than, e.g. a logistic regression, the resulting models are harder to interpret, and because the methods here are somewhat ad hoc, it is tough to get a sense for how the decisions the authors made in processing the data ended up affecting the results.

1a) In particular, the analyses involve a lot of comparisons of heavily derived summary statistics from random forest models, like AUC and feature importance, but the statistical significance of these differences is not always clear. While Figure 1 does appear to use a statistical test to evaluate AUC differences (for models trained on one age group and evaluated on the same/different age groups), the replication for CRC in Figure 3A does not. I also can't find a description of what test was used and what cutoff was called significant.

1b) The authors also say that the pattern they observe in Figure 3A is "exactly the same" as in Figure 1D, but it is difficult to tell whether this is accurate. In Figure 1D, performance mainly drops for models tested on elderly (E) samples, whereas in Figure 3A, performance mainly drops for models trained on (E) samples, with very little difference in evaluation performance when models trained on young/middle (YM) samples are evaluated on E samples. In fact, the AUCs in the lower panel of Figure 3A are all pretty similar, despite the strong color differences – a more globally consistent color scheme would help, and speaking of color, there seems to be an error in Figure 1D – Cirrhosis (0.97 is shaded darker than 0.95).

1c) In Figure 2, the cutoff (eighty fifth percentile of Gini decrease) is never justified and the subsequent filtering step struck me as unclear. The authors say they performed Mann Whitney U tests of marker scores across at least two age groups, but isn't there only one marker score per age bracket?

2) The authors mainly graph derived measures like AUC and feature importance without ever really visualizing the original data, with the exception of some of the metabolite measurements in the last figure. The closest we get are the name colors in Figure 2. This is important to sanity-check the results based on summary statistics, to show more clearly the extent of the age confounding, and to show what it is specifically about the aging-related markers that is being captured by the model. Visualization could capture whether, for example, in addition to greater loss, E samples had more "noise" or β-diversity compared to YM samples, and whether the aging signal was consistent across continents or driven by different taxa in different locations. Without visualizing, it's hard to get a sense of the scope of the issue.

3) The authors were sensitive to the fact that DNA sequencing/ extraction methodology can affect microbiome profiles which is good. They handled this by excluded some samples from a particular DNA extraction methodology that produced differences in observed diversity. It is strange, however, that after this step to reduce this confounding, addition samples from studies of IBD, CRC and ELDERMET were then added. Why would one add new samples after and not before taking steps to address effects of DNA sequencing/ extraction methodology? Later in the paper it becomes clearer that these other added studies are more validation cohorts and perhaps this is why? This should be clearer. Also, since there still was a minor effect of these factors after removing the samples with a large DNA extraction effect, it might be good to adjust for these differences in methods in the Adonis tests (e.g. DNA extraction + age in the model). Also, is seems that other factors about experimental protocol could also affect observed diversity- e.g. which kit was used to prepare the libraries (Tru-Seq versus others)?

4) It is unclear how the Adonis/PERMANOVA was performed to produce the data show in in Figure 1A. Our impression is that these tests were run independently for each metadata category but that is not completely clear. It seems that taking advantage of being able to control for confounding by running Adonis with more complex models (as suggested above for DNA extraction) might be good.

5) We'd like to see the data presentation and interpretation modified to try to avoid a potential bias towards age as an important covariate. Multiple metrics presented throughout the study are equivocal or even supportive of consistent trends despite age. The authors should be careful to point this out and to present a more nuanced interpretation that only some aspects of the microbiome are associated with aging.

6) The authors need to revise the main text to be more careful about the language used to avoid conflating correlation with causation. Words like "impact", "reinforced by", "influence", etc. imply that there is causal information where none exists.

7) The full dataset analyzed here needs to be made publicly available prior to publication. It's not sufficient to ask readers to re-assemble the full meta-analysis on their own.

8) Add analyses wherein each individual study is analyzed separately, as opposed to the current analysis which involves merging data across all the studies. The current approach has the downside of being influenced by the larger studies, whereas the former would allow for statements about the degree to which associations with age are reproducible across studies. That said, I'm not sure this is possible given the nature of the data. At a minimum, it would help to have some supplemental figures that depict the distribution of ages within and across studies.

9) The Discussion section really nails a critical confounder, drug use, which could explain a lot of what is described here. Diet and lifestyle are also likely contributors to these associations. While I don't think these data can be used to de-couple these factors, it would be good to be more explicit about this issue in the abstract and introduction. As written, the reader might take away the possibly false conclusion that aging itself (independent of diet, drugs, etc) is associated with the gut microbiome.

[Editors' note: further revisions were suggested prior to acceptance, as described below.]

Thank you for submitting your article "Adjusting for age improves identification of gut microbiome alterations in multiple diseases" for consideration by *eLife*. Your article has been reviewed by three peer reviewers, including Peter Turnbaugh as the Reviewing Editor and Reviewer #2, and the evaluation has been overseen by Wendy Garrett as the Senior Editor. The following individuals involved in review of your submission have agreed to reveal their identity: Catherine A Lozupone (Reviewer #1).

The reviewers have discussed the reviews with one another and the Reviewing Editor has drafted this decision to help you prepare a revised submission.

Summary:

The authors have expanded the analyses they originally put forth showing that age is a potential confounder in microbiome association studies. The addition of the PERMANOVA and the linear modeling makes the paper stronger, and the reviewers appreciated the authors making it clearer how they determined significance in the random forest studies.

Essential revisions:

Two remaining concerns about the statistic used need to be addressed prior to publication.

1) We appreciate seeing the distributions of AUCs that the authors generated, and this does help to make their point more convincing. However, now that we understand better what they did, we're not sure that the test they used is appropriate. It seems that the authors are partitioning studies into age groups, training on a randomly-sampled subset of one age group, then testing on the remaining held-out samples from the 1-2 other age groups. This is all good practice so far. The issue is that the authors seem to be doing this repeatedly to get a distribution of AUCs per cohort, then testing for significant differences between the AUC distributions per cohort, using a Mann-Whitney U test. The problem with this is that AUCs within a cohort are not actually independent of each other (which the M-W test requires), because some classifiers will have been trained on the same subsamples. This could make the distribution of AUCs look narrower than it really is. Further, the same classifier is being used on each age group, which means there are also correlated errors *across* cohorts. Testing for a significant difference between AUCs is actually a pretty subtle and difficult problem, which is another reason we prefer making statements about significance using explicit statistical models like logistic regression, whose properties are better understood.

If the authors really want to test for significant differences between these models, we would encourage them instead to use a permutation test: i.e., shuffle the labels of the *age cohorts* repeatedly, then use the empirical distribution of *differences* in AUCs between cohorts to test for significance. This should at least ensure that the correlations introduced by sampling and by testing the same classifier on multiple cohorts will also be present in the null distributions.

2) We also have one more substantive concern with Figure 5. We appreciate the authors did not drop low nominal p-values, but it is also not appropriate to use a fold-change cutoff *before* applying p-value corrections. Any filtering that uses information about the difference between groups (which includes fold-change cutoffs) is statistical double-dipping and will tend to inflate significance. You can either apply the fold-change cutoff *afte*r p-value correction or apply a filter that doesn't depend on knowledge about which samples belong to which groups, like the overall variance or the number of non-zero observations.

[Editors' note: further revisions were suggested prior to acceptance, as described below.]

Thank you for resubmitting your work entitled "Adjusting for age improves identification of gut microbiome alterations in multiple diseases" for further consideration by *eLife*. Your revised article has been evaluated by Wendy Garrett (Senior Editor) and a Reviewing Editor.

The manuscript has been improved but there are some remaining issues that need to be addressed before acceptance, as outlined below:

It is unclear in the main text why 2 different approaches are needed – especially since the last paragraph of subsection “Influence of age on the microbiome and microbiome-disease signatures” only refers to the results of the permutation based strategy. Seems like might be more straightforward to just report the latter, but at the least the results of both and not just one of the approaches should be reported if both are described.

In subsection “Influence of age on the microbiome and microbiome-disease signatures”, where authors say "In summary, in each disease-age-group scenario, we used two different approaches. " They should qualify for what – "e.g. we used two different approaches to assess whether disease classification performance was significantly different between same age group classification and different age group classification".

In subsection “Influence of age on the microbiome and microbiome-disease signatures”: "is significantly different from what would be expected by random (null distribution) " should briefly quality "generated by permuting the age-group labels of the subjects"

There was also a minor concern with the permutation test. The authors seem to be getting one average true or permuted AUC for each of the 100 resampled classifiers, instead of averaging across classifiers and getting one AUC per permutation (comparing the "true" value to this null to get a p-value). Because of non-independence between the training and test samples the classifiers used, the author's procedure should yield narrower AUC distributions, which could inflate significance. A back of the envelope estimate suggests an average overlap between classifiers of around 10-20% of the training + test samples. Given the consistency of the story and the magnitude of the differences the authors observe, though, as well as the computational time cost of doing enough permutations to get an accurate p-value, we leave it up to the authors whether they want to 'inoculate themselves against' this particular criticism.

Several different FDR cutoffs are used (0.15, 0.20, 0.25) with no explanation. Figure 6D seems particularly arbitrary in its use of two different thresholds, neither of which is that common (0.15 and 0.20). For that figure, in the absence of a really good reason to pick those specific cutoffs, it would probably be more transparent to just pick one standard threshold (e.g. 0.25) and to just make it really clear that Figure 6D represents a selection of the most significant of the resulting hits. The authors would of course also need to report how many total hits were significant at this threshold, and to make the full list available.

The authors should pick either one of Spearman or Kendall distances; using both in different places is confusing. (Bray-Curtis is another option for calculating divergence between microbiome abundance profiles.)

---

## [Author Response]

Essential revisions:1) The evaluation relies on random forests, a tool for classification, when the paper is really more about adjusting for confounders, which RFs don't intrinsically handle. The authors might be better served using an established meta-analytic framework like a mixed effects model. In order to use the RF for their purposes, the authors have to subsample their data extensively and have to perform a lot of pairwise analyses where controls from the same continent are pooled (and differences between countries on the same continent ignored). A hierarchical model should be able to capture and adjust for effects of continent and country in a way that made more efficient use of the available samples, would offer some protection against unbalanced examples, and would have more easily interpretable coefficients. While it's true that random forests have better classifier performance than, e.g. a logistic regression, the resulting models are harder to interpret, and because the methods here are somewhat ad hoc, it is tough to get a sense for how the decisions the authors made in processing the data ended up affecting the results.

We are grateful to the reviewers for this suggestion and have we have now addressed this as described below.

1) First, to establish that age as a covariate influences disease signature, the additional PERMANOVA has been performed for each disease with a more complex approach modelling the gut microbiome as a function of the interaction of age-group and disease taking country as a confounder as described below:

adonis(Microbiome ~ Country + Disease*Age-group)

This computes the extent of age-group associated disease signatures for each age-group (Disease:Age-group term) after adjusting for the extent of influence that disease and age-group independently have on the microbiome. For four of the diseases, age-group associated disease signatures (Disease:Age-group) has a significant influence on the microbiome composition (even after stratification for country). The newly added methodology used for this purpose has been described in subsection “Creation of generic disease prediction models and identification of shared disease markers”. The new results obtained in this analysis has been described in subsection “Influence of age on the microbiome and microbiome-disease signatures”. We have added a new Table 1 to show the results of this analysis. Overall, interaction between two metadata types measures the extent to which the microbiome variations with respect to one metadata (in this case, the disease) is influenced by the variation in the other. For three of the diseases (IBD, CRC and Polyps), the effect of the interaction of disease with age-group was significant (all with P < 0.05; PERMANOVA). The effect was also marginally significant for T2D (P < 0.08; PERMANOVA). This indicated that (even after taking into account the regional variations and the individual influences of disease and age), for these diseases, the microbiome variation associated with disease was significantly impacted by the variations in the age-group of the individuals. Further investigation of these influences (measured by the R^2^) also indicated that the extent of these effects was disease-specific, whereby IBD and CRC had the highest influence of age as a covariate of disease signatures, with T2D and Polyps having considerably lower influences (Table 1).

2) The second objective of this study was to probe and identify the taxa showing age-specific associations with disease. This is synonymous to asking to what extent age as a covariate influences the abundance changes associated with each taxon in different diseases. This was investigated using a combination of two different analogous approaches, as described below.

The first approach involved identifying the disease-associated microbiome signatures for different age-groups and investigating whether these signatures obtained from one age-group are extrapolatable to other age-groups, and subsequently investigating the differentially associated features. This was the approach that we had used in the previously submitted version of the manuscript.

We used Random Forest based approach because of the following reason. Random Forests can not only be used to judge the strength of the association between the predictors and the response (based on the accuracy or area under the curve (AUC) of prediction), but also whether associations identified on one set of observations are extrapolatable to another. Consequently, Random Forest models have been routinely used in microbiome based studies to not only quantify and characterize the microbiome associations with various diseases (Feng et al., 2015; Karlsson et al., 2013; Pasolli et al., 2016; Qin et al., 2012), but also to study the transferability of the associations predicted in set of individuals on others (He et al., 2018; Thomas et al., 2019). We have clarified this in subsection “Influence of age on the microbiome and microbiome-disease signatures”.

For such an approach, Random Forest, besides providing classification models superior to other methods like logistic regression, also provides the additional benefit of identifying the optimal features involved in the classification (as feature importance scores). The iterative decision tree-based approach implemented in Random Forests facilitates identification of co-occurrence. Another advantage is that the microbiome features (as a whole) could be inputted and used to create the model (something that cannot be performed in regression-based models). Additionally, to avoid overfitting and to ensure that the differences across age-groups were not a simple consequence of differences in the number of diseased and control individuals, we used an iterative approach (based on sub-sampling) that ensured uniformity of training and test sizes for each disease across all age-groups and the changes in feature signatures are not due to presence of specific unbalanced outlier examples for certain disease-age-group scenario.

However, we agree with the reviewers that, despite its simplicity, the above approach has one major limitation. It cannot consider the effect of confounders like (in this case), the country. We tried to address this first by restricting the Random Forest analysis to country-level bins (rather than continent-level bins as used in the previous submission) specific for each disease. Given that country had the highest effect on the gut microbiome, this was expected to address the regional confounding effect even more as compared to the previous version of the manuscript (as all comparisons are performed considering only those samples belonging to the same country as disease cohorts). The number of diseased and control samples obtained after this grouping has been summarized in a newly added Figure 1—figure supplement 3A. This grouping was not something we had explored in the initial version). This is described in subsection “Influence of age on the microbiome and microbiome-disease signatures”. By collating samples from only those datasets obtained from the same countries as the disease cohorts, we ensured that each disease-specific bin was regionally homogenous in terms of membership (thereby addressing the reviewer’s concern).

However, the above approach still does not capture one aspect. Microbiome associations with disease have both an age-group-specific and an age-group-independent component. However, many of the age-group-specific changes in microbiome-disease association may also be reflections of changes that accompany ageing in general. Age had the second major effect on microbiome composition, and it was important to investigate these age-specific changes in disease-microbiome associations after deconvoluting for the effect of ageing. Secondly, while grouping samples into country-level bins ensured enough regional homogeneity of the compared groups, there were still biases in the representation of certain regions in certain age-groups for the different diseases (Figure 1—figure supplement 3). Given that RF models cannot intrinsically adjust for these confounding effects, we probed this using a hierarchical linear regression-based strategy that compared the extent of influence that age-group had on the disease-association pattern of a taxon (Disease:Age-group) as described below.

In this linear model based validation strategy, within the disease-specific country-level bins, we modelled the (log transformed to the base 10) abundance of each species first as a function of disease-status and country and then as a function of the interaction of disease-status and age-group and country as an independent predictor as described below:

Model 1

Log(Taxon) ~ Country + Disease + Age-group

Model 2

Log(Taxon) ~ Country + Disease * Age-group, which is equivalent to:

Log(Taxon) ~ Country + Disease + Age-group + Disease:Age-group

The goodness of fit of both models was quantified using adjusted R-squares and AIC values, and the significance of improvement of Model 2 with respect to Model 1, was judged using one-sided log-likelihood tests. This took care of the country as a confounder as well as provided the extent to which the interaction of disease and age-group had an impact on the abundance of the species as compared to the individual factors.

We specifically investigated those taxa having significant differences in their feature importance scores (Figure 2—source data 1) and identified those having significantly higher influence of Disease interacting with Age-group than Age-group alone (Log likelihood test one sided P < 0.05) (Figure 2—source data 2; Figure 2B) as the final validated list of ‘strictly age-specific disease markers’. The results of this analysis have been described in subsection “Age-centric differences in microbiome-disease associations”. The methodology has been described in subsection “Validation of the shortlisted disease-associated taxonomic markers using Linear Models and identifying overlaps with taxa reported in original studies”.

3) The third key aspect of the current study was the gain versus loss patterns in disease association across age-groups. For this, we first performed the analyses within cohorts (now summarized in the new Figure 1C). We observed that for most of the diseases, the pattern obtained was like the one obtained earlier that is increase of taxa that are lost in disease in the elderly as compared to the young age-groups (newly added Figure 4—figure supplement 1). However, because of lower sample sizes and lower power, we had to perform the comparisons at relaxed thresholds of Mann-Whitney Test P-value < 0.05. Therefore, to increase statistical power and validate the above trends on a much larger cohort we repeated the analysis on a much larger continent-level bins and reproduce this pattern which was already observed in Figure 4A (unchanged in the current version). In this comparison, the features were identified with Benjamini Hochberg corrected Mann Whitney P < 0.1. These results have now been described in subsection “Reproducible association of the G1 disease-positive markers with increased frailty in elderly individuals from the ELDERMET cohort”.

In summary, in the current version of the manuscript, although cohort specific biases still exist, we have used a combination of Random Forest, linear models as well as validations at different levels of regional homogeneity to enhance the reliability of results as well as remove convoluting effect of general ageing on these age-specific disease signatures. We have described this in the Discussion section.

1a) In particular, the analyses involve a lot of comparisons of heavily derived summary statistics from random forest models, like AUC and feature importance, but the statistical significance of these differences is not always clear. While Figure 1 does appear to use a statistical test to evaluate AUC differences (for models trained on one age group and evaluated on the same/different age groups), the replication for CRC in Figure 3A does not. I also can't find a description of what test was used and what cutoff was called significant.

For both Figure 2A (earlier Figure 1D) and 3A, we have used Mann-Whitney U tests to compare the AUC distributions for the 100 Random Forest iterations obtained for the various scenario. Instead of tables, Figure 2A now shows the boxplot of the distribution of disease classification AUCs of classifiers trained on either the same or different age-groups. We feel that this clearly shows the differences in the classification performances across age-groups, and removes the confusion associated with the colours and values as earlier. We have now described this in the revised manuscript (subsection “Age-centric differences in microbiome-disease associations”; subsection “Age-specific changes in the directionality of taxon abundance alterations for specific diseases, and the microbiome response shared by multiple diseases”).

1b) The authors also say that the pattern they observe in Figure 3A is "exactly the same" as in Figure 1D, but it is difficult to tell whether this is accurate. In Figure 1D, performance mainly drops for models tested on elderly (E) samples, whereas in Figure 3A, performance mainly drops for models trained on (E) samples, with very little difference in evaluation performance when models trained on young/middle (YM) samples are evaluated on E samples. In fact, the AUCs in the lower panel of Figure 3A are all pretty similar, despite the strong color differences – a more globally consistent color scheme would help, and speaking of color, there seems to be an error in Figure 1D – Cirrhosis (0.97 is shaded darker than 0.95).

We apologize for this lack of precision in our language. Our intention was to convey that the pattern observed in Figure 3A was similar to that observed in the earlier Figure 1D (now Figure 2A) for CRC, where in the highest AUC was observed for classifiers trained on the young and the middle aged group and tested on individuals belonging to the same age-group, with a significant loss of performance for those either trained or tested on the elderly individuals. This indicates a loss of microbiome associated disease signature for CRC in the elderly. To avoid confusion arising out of colour differences in heatmaps showing median values of AUC (in both the earlier Figure 1D and Figure 3A), we have now provided boxplots that show the distribution as well as the comparison of AUC values using Mann-Whitney Tests. We have also improved the language used to describe the similarity of the age/AUC correlations (subsection “Age-specific changes in the directionality of taxon abundance alterations for specific diseases, and the microbiome response shared by multiple diseases”).

1c) In Figure 2, the cutoff (eighty fifth percentile of Gini decrease) is never justified and the subsequent filtering step struck me as unclear. The authors say they performed Mann Whitney U tests of marker scores across at least two age groups, but isn't there only one marker score per age bracket?

To clarify this reviewer comment, a short explanation of how we arrived at this eighty fifth percentile threshold has now been added to the revised version of the manuscript (Materials and methods section) along with an additional Figure 2—figure supplement 1. The explanation is summarized below:

The iterative sub-sampling based Random Forest analysis avoided overfitting and, by virtue of the uniformity of training and test sizes for each disease across all age-groups, ensured that the differences across age-groups were not a simple consequence of differences in the number of diseased and control individuals. The objective of the iterative approach was also to identify taxa that showed stable association with disease irrespective of the sub-sampling population. This aimed to identify taxa that were always considered for the classification model irrespective of the sampled population of diseased and controls (Mean Decrease of GINI of at least 0 across at least 95% of iterations). So, for each disease/age-group scenario, we aimed to identify that minimum percentile threshold above which the taxa were considered for the classification model across at least 95% of the iterations (Figure 2—figure supplement 1A). We observed that for taxa, having a percentile score of at least 85, included in training models across at least 95% of the iterations across all the 13 disease age-group scenarios. Further plotting the mean feature scores of taxa arranged in terms of their percentile scores, we observed that the mean feature importance scores remained stable and low till the eightieth percentile mark and started increasing considerably only after that (Figure 2—figure supplement 1B). Based on these two results, the taxa having importance scores in the top 15 percentile (that is higher than eighty fifth percentile) were identified as the top eighty fifth percentile predictors/markers for a given disease in that age-group.

2) The authors mainly graph derived measures like AUC and feature importance without ever really visualizing the original data, with the exception of some of the metabolite measurements in the last figure. The closest we get are the name colors in Figure 2. This is important to sanity-check the results based on summary statistics, to show more clearly the extent of the age confounding, and to show what it is specifically about the aging-related markers that is being captured by the model. Visualization could capture whether, for example, in addition to greater loss, E samples had more "noise" or β-diversity compared to YM samples, and whether the aging signal was consistent across continents or driven by different taxa in different locations. Without visualizing, it's hard to get a sense of the scope of the issue.

We thank the reviewers for pointing this out. As described above in responding to previous comments, we have now performed additional analysis in the current version of the manuscript to address these concerns. To identify the extent of age-confounding, using PERMANOVA analysis, we first evaluate and establish whether and to extent age as a covariate influences taking into account the country-specific variations in microbiome signatures as described below:

adonis(Microbiome ~ Country + Disease*Age-group)

This computes the extent of influence that disease and age-group independently have on the microbiome as well as the extent of age-group associated disease signatures for each age-group (Disease:Age-group term). For four of the diseases, age-group associated disease signatures (Disease:Age-group) has a significant influence on the microbiome composition (even after stratification for country). The newly added methodology used for this purpose has been described in subsection “Creation of generic disease prediction models and identification of shared disease markers”. The new results obtained in this analysis has been described in subsection “Influence of age on the microbiome and microbiome-disease signatures”. We have added a new Table 1 to show the results of this analysis.

Regarding the second point of the reviewer regarding the nature of ageing related markers that were being identified by this approach. Age-group specific markers can be classified in two types. The first type includes those that show differential associations with disease, because their abundance changes with increasing age (even in the controls). The second type includes those which show significantly different associations with disease across age-groups even after accounting for the changes that happen with ageing. In the revised version of the manuscript, we have used a combination of both approaches, namely Random Forest and Linear Modelling to understand these effects. RF would identify both types. the modelling would identify the second. Consequently, we first use Random Forest models to assess the performance of disease prediction models devised on one age-group on others as well as to identify a filtered optimal list of age-group specific markers (as earlier). Subsequently, we validate the age-specific disease association trends of each, filtered using linear models adjusting for the effect of country-specific variations. In this validation approach, we model the (centralized log ratio transformed) abundance of each species, first as a function of disease-status and country, and then as a function of the interaction of disease-status and age-group and country as an independent predictor, as described below:

Model 1

Log(Species) ~ Country + Disease + Age-group

Model 2:

Log(Species) ~ Country + Disease * Age-group

The goodness of fit of both models was quantified using adjusted R-squares and AIC values, and the significance of improvement of Model 2 with respect to Model 1, was judged using log-likelihood tests. This took care of the country as a confounder as well as provided the extent to which the interaction of disease and age-group had an impact on the abundance of the species as compared to the individual factors.

We specifically investigated those taxa having significant differences in their feature importance scores (Figure 2—source data 1) and identified those having significantly higher influence of Disease interacting with Age-group than Age-group alone (Log likelihood test one sided P < 0.05) (Figure 2—source data 2; Figure 2B) as the final validated list of ‘strictly age-specific disease markers’. The results of this analysis have been described in subsection “Age-centric differences in microbiome-disease associations”. The methodology has been described in subsection “Validation of the shortlisted disease-associated taxonomic markers using Linear Models and identifying overlaps with taxa reported in original studies”.

Furthermore, in line with the reviewers’ suggestion, we also checked whether the samples from elderly controls had significantly higher variability as compared to young/middle-aged controls, across the different continental regions (Figure 4—figure supplement 2). In line with our hypothesis, for both Europe and North America, gut microbiome from elderly individuals were significantly more variable as compared to young and middle-aged controls. This was however not observed in the Asia (Chinese) cohort. Interestingly, Cirrhosis which neither shows neither an association with disease signatures nor the gain versus loss pattern contained only subjects from Asia (summarized in subsection “Reproducible association of the G1 disease-positive markers with increased frailty in elderly individuals from the ELDERMET cohort”). This indicates that ageing has different patterns across continents. This increasing loss could lead to dysbiotic configurations characterized by higher inter individual variability, increased abundance of pathobionts (resulting in loss of disease signature) thereby making the microbiome more susceptible to diseases.

However, due to skewness of the age distribution across datasets, validations with respect to the reproducibility of the disease signatures could not be performed for all diseases. However, for CRC (which was observed to have one of the strongest influences of age on disease signatures), we had additional validation cohorts and we have tested these aspects extensively in the validation cohorts (Figure 3; subsection “Age-specific changes in the directionality of taxon abundance alterations for specific diseases, and the microbiome response shared by multiple diseases”).

3) The authors were sensitive to the fact that DNA sequencing/ extraction methodology can affect microbiome profiles which is good. They handled this by excluded some samples from a particular DNA extraction methodology that produced differences in observed diversity. It is strange, however, that after this step to reduce this confounding, addition samples from studies of IBD, CRC and ELDERMET were then added. Why would one add new samples after and not before taking steps to address effects of DNA sequencing/ extraction methodology? Later in the paper it becomes clearer that these other added studies are more validation cohorts and perhaps this is why? This should be clearer. Also, since there still was a minor effect of these factors after removing the samples with a large DNA extraction effect, it might be good to adjust for these differences in methods in the Adonis tests (e.g. DNA extraction + age in the model). Also, is seems that other factors about experimental protocol could also affect observed diversity- e.g. which kit was used to prepare the libraries (Tru-Seq versus others)?

As pointed out by the reviewer, the additional samples were added as validation cohorts. We have clarified this in subsection “Influence of age on the microbiome and microbiome-disease signatures”. We also appreciate the reviewers’ suggestion to include the DNA extraction method as a factor and have included this in the computation of the PERMANOVA values in Figure 1A,B of the current version of the manuscript, as noted in subsection “Influence of age on the microbiome and microbiome-disease signatures” (described also in Materials and methods section).

4) It is unclear how the Adonis/PERMANOVA was performed to produce the data show in in Figure 1A. Our impression is that these tests were run independently for each metadata category but that is not completely clear. It seems that taking advantage of being able to control for confounding by running Adonis with more complex models (as suggested above for DNA extraction) might be good.

As suggested by the reviewer, we have now included the DNA extraction method as a factor in the PERMANOVA analysis for Figure 1A, and briefly mentioned this in subsection “Influence of age on the microbiome and microbiome-disease signatures” and in the Materials and methods section. However, we would like to respectfully point out that the objective of this analysis was to independently assess the importance of each of the factors on the microbiome composition and not to add complexity to the model. This analysis provided us the relative importance of each metadata on the microbiome composition, based on which we have addressed the appropriate confounders in the subsequent analyses.

5) We'd like to see the data presentation and interpretation modified to try to avoid a potential bias towards age as an important covariate. Multiple metrics presented throughout the study are equivocal or even supportive of consistent trends despite age. The authors should be careful to point this out and to present a more nuanced interpretation that only some aspects of the microbiome are associated with aging.

We appreciate the reviewer’s suggestion. The PERMANOVA analysis performed in the current version precisely addresses this. Age as a covariate of disease signature has a significant but reduced effect on microbiome composition. The extent of these associations is also disease specific. We have explained this in subsection “Influence of age on the microbiome and microbiome-disease signatures” and the Materials and methods section. We have also modified the Results section to remove ‘strong’ words in the text. Wherever appropriate, we have pointed out possible confounders and limitations of the current study in the Discussions section.

6) The authors need to revise the main text to be more careful about the language used to avoid conflating correlation with causation. Words like "impact", "reinforced by", "influence", etc. imply that there is causal information where none exists.

As mentioned in the previous response, we have made our results narrative much more pragmatic in the current version of the manuscript, for example in the Introduction.

7) The full dataset analyzed here needs to be made publicly available prior to publication. It's not sufficient to ask readers to re-assemble the full meta-analysis on their own.

The key in-house source codes used in this meta-analysis have been provided as Supplementary file 4. The shotgun data of the ELDERMET is available for download from the ELDERMET website at http://eldermet.ucc.ie/temp1/eldermet_shotgun_data_filtered_all_sample.tar. We have now included this database information in the data availability statement of the manuscript.

8) Add analyses wherein each individual study is analyzed separately, as opposed to the current analysis which involves merging data across all the studies. The current approach has the downside of being influenced by the larger studies, whereas the former would allow for statements about the degree to which associations with age are reproducible across studies. That said, I'm not sure this is possible given the nature of the data. At a minimum, it would help to have some supplemental figures that depict the distribution of ages within and across studies.

We appreciate the reviewers’ suggestion to perform the analyses within individual study cohorts to show which results are reproduced within cohorts and those that are cohort specific. However, as pointed by the reviewers, and shown in Figure 1C and 1E, each of the individual cohorts has biases for samples belonging to certain age-groups. Four of the eight datasets analysed in the study have a significant variation in the age of diseased and control individuals. However, for CRC, we had additional validation cohorts and we have tested these aspects extensively in the validation cohorts (Figure 3; subsection “Age-specific changes in the directionality of taxon abundance alterations for specific diseases, and the microbiome response shared by multiple diseases”).

9) The Discussion section really nails a critical confounder, drug use, which could explain a lot of what is described here. Diet and lifestyle are also likely contributors to these associations. While I don't think these data can be used to de-couple these factors, it would be good to be more explicit about this issue in the abstract and introduction. As written, the reader might take away the possibly false conclusion that aging itself (independent of diet, drugs, etc) is associated with the gut microbiome.

We appreciate the reviewers’ suggestion and have added appropriate statements in the Introduction that describe these aspects. Specifically, we have mentioned that these metadata that can have an influence on gut microbiome but not available in curatedMetagenomicData (Introduction).

[Editors' note: further revisions were suggested prior to acceptance, as described below.]

Essential revisions:Two remaining concerns about the statistic used need to be addressed prior to publication.1) We appreciate seeing the distributions of AUCs that the authors generated, and this does help to make their point more convincing. However, now that we understand better what they did, we're not sure that the test they used is appropriate. It seems that the authors are partitioning studies into age groups, training on a randomly-sampled subset of one age group, then testing on the remaining held-out samples from the 1-2 other age groups. This is all good practice so far. The issue is that the authors seem to be doing this repeatedly to get a distribution of AUCs per cohort, then testing for significant differences between the AUC distributions per cohort, using a Mann-Whitney U test. The problem with this is that AUCs within a cohort are not actually independent of each other (which the M-W test requires), because some classifiers will have been trained on the same subsamples. This could make the distribution of AUCs look narrower than it really is. Further, the same classifier is being used on each age group, which means there are also correlated errors across cohorts. Testing for a significant difference between AUCs is actually a pretty subtle and difficult problem, which is another reason we prefer making statements about significance using explicit statistical models like logistic regression, whose properties are better understood.If the authors really want to test for significant differences between these models, we would encourage them instead to use a permutation test: i.e., shuffle the labels of the age cohorts repeatedly, then use the empirical distribution of differences in AUCs between cohorts to test for significance. This should at least ensure that the correlations introduced by sampling and by testing the same classifier on multiple cohorts will also be present in the null distributions.

While we appreciate the reviewer’s thoughtful response and we agree that the AUCs are not truly independent and so any statistical test will give a P-value that is lower than if we had many truly independent (non-overlapping) cohorts, this lack of true independence would only be problematic in a situation where the observed biological (size-) effect is small and replication is not seen across country and disease cohorts. However, we have replicated these findings across multiple datasets and have observed large effect sizes (0.04 – 0.38 difference in AUC). Even though errors can be correlated across age cohorts, it is unlikely that they are correlated across country and disease cohorts; we still observe the difference in same age-group classification versus different age-group classification. Furthermore, if the analysis was affected by a number of ‘difficult to classify’ samples, the difficult to classify samples would be difficult to classify independently of the prediction model used, and so we would not expect it to lead to the observed differences.

However, in line with the reviewers’ suggestions, we have modified the strategy used to test whether our RF-based models show significant age-group-specific variations in their prediction accuracy for the various diseases, using two different approaches.

In the first approach, we created two ‘re-sampled’ tests sets. While one contained Y diseased and Y control individuals from the same age-group (but not included in the training sub-set) (Same Age-group test set), the other contained Y diseased and Y control individuals from the other two age-groups (Different Age-group test set) (with Y defined as in Figure 2—source data 1). We tested the classifier on each test set and computed the AUCs. Testing the same classifier on both test sets ensured that the observed variations in disease prediction performance was not due to differences in the subject sub-samples used to create the classification models. Furthermore, to ensure that we don’t have biases introduced because of the selection of test sets (same age-group, different age-group), we repeated above steps 20 times (per classifier in each iteration) and computed the median AUCs for both the Same Age-group classification and the Different age-group classification. These median AUCs obtained for the Same Age-group and Different age-group classification for each of the 100 iterative sub-sampled classifiers were compared using Wilcoxon Signed Rank tests (to check if the performance of the classifiers significantly varied when tested on the same or different age-groups).

In the second approach, using Permutation test framework, we tested whether observed difference in classification AUCs (i.e. AUC for Same Age-group classification – AUC for Different Age-group classification) was significantly different than what would be expected by random chance. For this, we needed null distribution of empirical differences of AUC for the same sub-sampled classifier. For this purpose, for each of the 20 iterations corresponding to each of the 100 sub-sample based RF classifier models (as described in the previous paragraph), we first merged the two ‘re-sampled’ test sets (Same Age-group test set and Different Age-group test set), then permuted the age-group labels of the subjects, creating two ‘Permuted’ test tests (Permuted test set 1 and 2), tested the classifier model for each of the Permuted test sets, and finally computed the AUC differences obtained for the same classifier between the two permuted test sets. For each of the RF classifiers (across the 100 iterations), we computed the medians of the differences of AUCs (for the two permuted test sets) across the 20 iterations. The median difference of the AUCs obtained for the actual test (for Same Age-group – Different Age-group) and the permuted tests (Permuted Test set 1 – Permuted test set 2) obtained for the 100 iterations were then compared with Wilcoxon signed rank-tests. The objective of these permutation tests was to reduce the effects of correlated errors associated with certain ‘aberrant’ samples in influencing the disease prediction performance of RF classifiers for specific age-groups. This permutation test is expected at least ensure that the correlations introduced by sampling and by testing the same classifier on multiple cohorts will also be present in the null distributions. For both the approaches, the p-values of comparison across age-groups were corrected using Holms’ correction.

These details have now been included in subsection “Disease classification using Random Forest (RF) Models”.

The Results section has also been modified accordingly and provided in Figure 2 (for the first approach) and Figure 2—figure supplement 2 (for the permutation test based approach).

The complete schematic summary of this approach has now been provided in Figure 2—figure supplement 1.

For uniformity, we have also used the same strategy for the validation of age-specific disease association trends in the CRC cohorts in Figure 4A (earlier Figure 3) and newly added Figure 4—figure supplement 1. These results have been described in subsection “Replicability of the age-centric microbiome-disease signatures across multiple cohorts in CRC2”.

We have also additional source codes for this analysis that can be tested on the cohorts from the curatedMetagenomicData as well as additional in-house cohorts.

2) We also have one more substantive concern with Figure 5. We appreciate the authors did not drop low nominal p-values, but it is also not appropriate to use a fold-change cutoff before applying p-value corrections. Any filtering that uses information about the difference between groups (which includes fold-change cutoffs) is statistical double-dipping and will tend to inflate significance. You can either apply the fold-change cutoff after p-value correction or apply a filter that doesn't depend on knowledge about which samples belong to which groups, like the overall variance or the number of non-zero observations.

We appreciate the reviewers’ concern and have now removed the fold-change cut-off. We have now modified our identification strategy to identify the frailty-associated metabolites. There are two specific properties of these metabolite profiles that were important:

These metabolite profiles should be enriched in their detection in the frailty associated taxonomic markers. But not all metabolite profiles associated with these markers may be associated with frailty.

These metabolite profiles should also independently show significant negative associations with Frailty scores (if we take the cumulative abundance of all species known in literature to have the given metabolite profile).

Thus, in the current manuscript version, we identified the frailty-linked metabolites associated with the eight taxonomic markers of frailty using a two-step strategy. First, we performed a correlation analysis of each metabolite profile (i.e. the cumulated abundance of taxa previously associated in literature with a given metabolic capability as obtained above) with FIM scores and identified metabolite profiles that showed significant association with FIM scores (Spearman Rho; FDR of less than 0.15). Next, we identified which of these identified metabolite profiles were detected in the taxonomic markers of frailty (based on previous literature) at a rate significantly higher than the background detection (using our Fishers’ exact test approach with FDR corrected P < 0.20.

This has been summarized in subsection “Creating metabolite species maps and obtaining the frailty-associated metabolic signature of a given group of species”.

In the first step, a total of 82 metabolite profiles were observed to have significant association with FIM scores (Spearman Rho; FDR of less than 0.15) (shown in newly added Figure 5—figure supplement 2). We observed that this association analysis of metabolite profiles reflects frailty-associated changes with respect to bioavailability of specific compounds, many of which have been previously shown to have corresponding associations with health, thereby corroborating our findings (Claesson et al., 2012). Specifically, the onset of frailty is observed to be associated with an increase of SCFA consumption by gut-bacteria (accompanied by a concomitant decrease of the production of the SCFA butyrate), increased consumption of the beneficial amino acid Tryptophan as well as increased production of the T2D-linked amino acid Threonine.

This has been summarized in subsection “Reproducible association of the G1 disease-positive markers with increased frailty in elderly individuals from the ELDERMET cohort”.

In the next step, we focussed on the group of frailty marker taxa and identified 13 metabolic profiles that were significantly associated with the eight frailty-marker taxa. This subset of 13 metabolite profiles alone could predict frailty with a R value of 0.60 (between the actual and predicted FIM values), which is significantly higher than that obtained after removing these features from the metabolite profile (subsection “Reproducible association of the G1 disease-positive markers with increased frailty in elderly individuals from the ELDERMET cohort”).

In summary, 12 of our earlier identified 16 metabolites are retained in this list. We don’t detect Acetate Production, Xylose Consumption, Dextrin Consumption and CO2 Production in this new list. However, the most important observation is the addition of Trimethylamine Production which agrees perfectly well with our detection of high Choline consumption, an increased abundance of CutC enzyme as well as further indicating the detrimental effects of these frailty-associated taxonomic markers. Figure 6 (earlier Figure 5) has now been modified to include these results.

[Editors' note: further revisions were suggested prior to acceptance, as described below.]

It is unclear in the main text why 2 different approaches are needed – especially since the last paragraph of subsection “Influence of age on the microbiome and microbiome-disease signatures” only refers to the results of the permutation based strategy. Seems like might be more straightforward to just report the latter, but at the least the results of both and not just one of the approaches should be reported if both are described.

We would like to note that we had already mentioned the results using both the approaches (Please refer to the last paragraph of subsection “Influence of age on the microbiome and microbiome-disease signatures”). This section reads as follows:

“The data revealed large differences across age groups for the various diseases. Specifically, in 10 of the 13 disease-age group scenarios (covering five diseases), classifiers trained and tested on the same age-group had significantly higher disease prediction AUC than when tested on different age-groups (Figure 2), with the improvement of classification performance significantly higher than would be expected by random chance (obtained using the permutation test based strategy) (Figure 2—figure supplement 2). This confirmed that microbiome-disease associations had age-centric trends.”

However, for the specific case of CRC and Polyps, we have now modified the latter half of subsection “Age-specific changes in the directionality of taxon abundance alterations for specific diseases, and the microbiome response shared by multiple diseases”. The section reads as below:

“CRC and Polyps were characterized by noticeably similar age-specific trends wherein the elderly age-groups had noticeable decrease in classification AUCs (that is lower AUCs for classifiers trained or tested on the elderly age-groups), indicating that the microbiome-disease signatures for these diseases in the elderly age-groups are weaker (Figure 2). For these two diseases, the AUC difference between the Same Age-group and Different Age-group disease classification was also reduced in the elderly (in the case of Polyps, this difference was not significantly greater that what would be expected by random chance) (Figure 2—figure supplement 2).”

In subsection “Influence of age on the microbiome and microbiome-disease signatures”, where authors say "In summary, in each disease-age-group scenario, we used two different approaches. " They should qualify for what – "e.g. we used two different approaches to assess whether disease classification performance was significantly different between same age group classification and different age group classification".

We have now clarified this.

In subsection “Influence of age on the microbiome and microbiome-disease signatures”: "is significantly different from what would be expected by random (null distribution) " should briefly quality "generated by permuting the age-group labels of the subjects".

We have now clarified.

There was also a minor concern with the permutation test. The authors seem to be getting one average true or permuted AUC for each of the 100 resampled classifiers, instead of averaging across classifiers and getting one AUC per permutation (comparing the "true" value to this null to get a p-value). Because of non-independence between the training and test samples the classifiers used, the author's procedure should yield narrower AUC distributions, which could inflate significance. A back of the envelope estimate suggests an average overlap between classifiers of around 10-20% of the training + test samples. Given the consistency of the story and the magnitude of the differences the authors observe, though, as well as the computational time cost of doing enough permutations to get an accurate p-value, we leave it up to the authors whether they want to 'inoculate themselves against' this particular criticism.

We have now explicitly mentioned and discussed this concern (as mentioned by the reviewer in the last paragraph) of the Discussions section as:

“The last concern is methodological and pertains to the non-independence between the training samples in the iterative RF classifiers generated for each disease-age-group scenario. Classifiers generated in each iteration are likely to share overlap of training (and/or testing) samples, thereby potentially resulting in narrower AUC distributions which could inflate significance values. However, the magnitude of the differences as well as consistency of the results (using multiple methods and study cohorts) clearly indicate the reliability of the results obtained in this study.”

We hope the clarification provided in the above paragraph addresses the reviewers’ concern regarding the over-interpretation of the results obtained using the methodology adopted in the current study.

Several different FDR cutoffs are used (0.15, 0.20, 0.25) with no explanation. Figure 6D seems particularly arbitrary in its use of two different thresholds, neither of which is that common (0.15 and 0.20). For that figure, in the absence of a really good reason to pick those specific cutoffs, it would probably be more transparent to just pick one standard threshold (e.g. 0.25) and to just make it really clear that Figure 6D represents a selection of the most significant of the resulting hits. The authors would of course also need to report how many total hits were significant at this threshold, and to make the full list available.

We have now adopted a single universal standard threshold of FDR < 0.25 for both stages of the identification of frailty-marker associated detrimental metabolites (Spearman Correlation associating the Metabolite Profiles with FIM values, followed by Fishers’ exact test analysis associating the FIM-associated Metabolite profiles identified in the Spearman Correlation analysis with the Frailty-associated taxonomic markers). Given this modification of the approach, the full list of metabolite profiles showing an association with FIM scores shows a slight change. Accordingly, Figure 6—figure supplement 2 that shows the full list of these metabolite profiles has been modified. However, the final list of 13 metabolite profiles shown in Figure 6D does not change even after increasing the threshold to FDR < 0.25.

The authors should pick either one of Spearman or Kendall distances; using both in different places is confusing. (Bray-Curtis is another option for calculating divergence between microbiome abundance profiles.)

We have now replaced Kendall distances, and used Spearman Distance measures instead, throughout the manuscript. Accordingly, boxplot of Figure 4C has been modified to show the Spearman distances of the feature rank profiles for different age-groups across the training cohorts.